# Peering into the Unknown: Active View Selection with Neural Uncertainty Maps for 3D Reconstruction

**Zhengquan Zhang**[1,2]    **Feng Xu**[2]    **Mengmi Zhang**[1†]
[1]Deep NeuroCognition Lab, Nanyang Technological University, Singapore
[2]Fudan University, China
[†] Address correspondence to `mengmi.zhang@ntu.edu.sg`

## Abstract

Imagine trying to understand the shape of a teapot by viewing it from the front—you might see the spout, but completely miss the handle. Some perspectives naturally provide more information than others. How can an AI system determine which viewpoint offers the most valuable insight for accurate and efficient 3D object reconstruction? Active view selection (AVS) for 3D reconstruction remains a fundamental challenge in computer vision. The aim is to identify the minimal set of views that yields the most accurate 3D reconstruction. Instead of learning radiance fields, like NeRF or 3D Gaussian Splatting, from a current observation and computing uncertainty for each candidate viewpoint, we introduce a novel AVS approach guided by neural uncertainty maps predicted by a lightweight feedforward deep neural network, named UPNet. UPNet takes a single input image of a 3D object and outputs a predicted uncertainty map, representing uncertainty values across all possible candidate viewpoints. By leveraging heuristics derived from observing many natural objects and their associated uncertainty patterns, we train UPNet to learn a direct mapping from viewpoint appearance to uncertainty in the underlying volumetric representations. Next, our approach aggregates all previously predicted neural uncertainty maps to suppress redundant candidate viewpoints and effectively select the most informative one. Using these selected viewpoints, we train 3D neural rendering models and evaluate the quality of novel view synthesis against other competitive AVS methods. Remarkably, despite using half of the viewpoints than the upper bound, our method achieves comparable reconstruction accuracy. In addition, it significantly reduces computational overhead during AVS, achieving up to a 400 times speedup along with over 50% reductions in CPU, RAM, and GPU usage compared to baseline methods. Notably, our approach generalizes effectively to AVS tasks involving novel object categories, without requiring any additional training. All code, models, and datasets are available at `https://github.com/ZhangLab-DeepNeuroCogLab/PUN`.

## 1 Introduction

Actively interacting with the environment to reduce uncertainty and minimize prediction errors is a fundamental capability of embodied intelligent systems Han & Zhang (2024); Friston (2010); Zhang & Xu (2024). Some viewpoints naturally offer more informative observations than others—for example, a front view of a teapot may only reveal the spout, providing limited information, whereas a side view can expose both the handle, the spout, and detailed surface textures of the body. Active View Selection (AVS) Sequeira et al. (1996); Jia et al. (2009); Connolly (1985); Pito (1999) addresses this problem by selecting a minimal set of viewpoints that collectively maximize information gain. See **Fig. 1(a)** for an illustration of the AVS task in the context of 3D object reconstruction. AVS is critical in a range of real-world applications, including robotic control Khandelwal et al. (2023); Zhang et al. (2018b); Lv et al. (2023), search and rescue Zhang et al.

(2022); Jia et al. (2025); Ding et al. (2022); Gupta et al. (2021); Wang et al. (2025); Niroui et al. (2019); Zhang et al. (2018a), and cultural heritage digitization Serafin et al. (2016).

Classical 3D reconstruction methods represent objects using explicit volumetric formats such as point clouds Dornhege & Kleiner (2013), voxel grids Dai et al. (2020); Dang et al. (2018), or occupancy maps Georgakis et al. (2022); Ramakrishnan et al. (2020). AVS methods built on these representations typically use depth images to estimate uncertainty in the 3D reconstruction, enabling efficient and real-time viewpoint selection. However, these approaches typically produce low-fidelity 3D reconstructions, due to their strong dependence on the quality of depth images.

Recently, neural rendering methods such as Neural Radiance Fields (NeRF) Yu et al. (2021); Wang et al. (2021); Trevithick & Yang (2021); Mildenhall et al. (2021) and 3D Gaussian Splatting (3DGS) Kerbl et al. (2023); Szymanowicz et al. (2024); Charatan et al. (2024); Chen et al. (2024) have gained significant attention for their strong performance in 3D object reconstruction. However, their effectiveness comes with the high computational cost, largely due to the large number of viewpoints required for training. To mitigate this issue, recent research has explored acceleration strategies such as few-shot training Jain et al. (2021), optimized ray sampling Li et al. (2023), and hybrid scene representations that combine continuous volumetric functions with voxel grids Cao et al. (2024).

Another promising direction is Active View Selection (AVS), which aims to identify and use only a small subset of the most informative viewpoints. Most existing AVS methods Pan et al. (2022); Ran et al. (2023); Lee et al. (2022); Yan et al. (2023); Zhan et al. (2022); Xue et al. (2024); Sünderhauf et al. (2023); Hoffman et al. (2023); Jin et al. (2023) follow a two-stage pipeline: they first train a neural rendering model using a limited set of views, then estimate the utility of candidate viewpoints using uncertainty heuristics derived from the trained model. The next view is selected based on these heuristics. For instance, several AVS approaches Pan et al. (2022); Ran et al. (2023); Lee et al. (2022); Yan et al. (2023); Zhan et al. (2022); Xue et al. (2024); Yu et al. (2021) train a NeRF model on the current set of views, and then evaluate the uncertainty of each candidate view. Some methods Lee et al. (2022); Zhan et al. (2022); Yan et al. (2023) estimate uncertainty by computing the entropy of opacity distributions along rays. Later approaches Pan et al. (2022); Ran et al. (2023); Xue et al. (2024) incorporate pixel-level reconstruction quality by averaging color variance across sampled points along each ray. Xue et al. (2024) further enhances this by factoring in visibility, assigning higher uncertainty to occluded or previously unobserved regions. However, these methods require retraining the NeRF model after each new view is added, resulting in significant computational overhead and slow iterative training cycles. As a result, current NeRF-based AVS pipelines are impractical for applications with limited time or computational resources. A group of existing approaches Jin et al. (2023); Smith et al. (2022) utilize pre-trained multi-view reconstruction models to avoid the costly retraining process. Specifically, Smith et al. (2022) employs a pre-trained reconstruction model at inference to predict occupancy from multiple input views, and then derives geometry-based uncertainty from the predicted occupancy. However, this is not an end-to-end pipeline: the model makes indirect predictions by first estimating occupancy and then converting it into uncertainty. This two-step procedure can be time-consuming, and errors in occupancy prediction may propagate and compound during uncertainty estimation. Recently, AVS methods have leveraged properties of 3D Gaussians from 3DGS models to estimate uncertainty. ActiveSplat Li et al. (2024) uses Gaussian variance to model scene density and selects viewpoints that maximize spatial coverage. ActiveGS Jin et al. (2025) and ActiveGAMER Chen et al. (2025) employ depth supervision and Gaussian density to estimate the number of currently occluded voxels that would be revealed from each candidate view. However, these methods primarily focus on geometry and overlook rich pixel-level cues such as color in the volumetric representations.

To address these limitations, another class of AVS approaches aims to improve computational efficiency by learning a direct mapping from the current viewpoint to the next optimal one. This is typically achieved via supervised learning Mendoza et al. (2020), using ground-truth next-best-viewpoints that maximize point cloud coverage, or through reinforcement learning Wang et al. (2024), where the reward is based on the increase in voxel or point cloud coverage after next view selection. However, these methods rely on a fixed, discrete set of candidate viewpoints , which makes them rigid and limits their generalizability to novel viewpoints in new environments.

Here, we propose a novel AVS method called PUN (Peering into the UnkNowN), which consists of two components: (1) neural uncertainty map prediction and (2) next-best-view selection. First, we introduce UPNet (Uncertainty Prediction Network), a lightweight feedforward neural network that

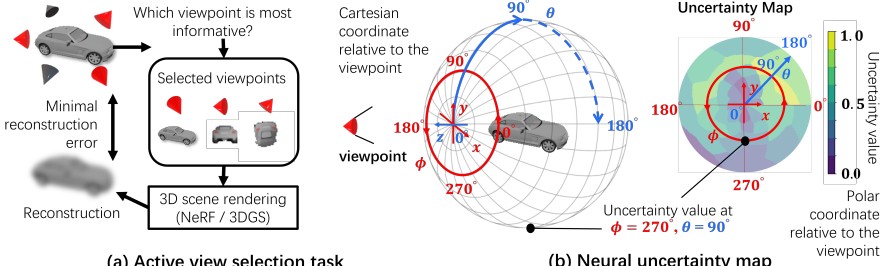

Figure 1: **(a) Illustration of Active Viewpoint Selection (AVS).** The goal of AVS is to select the most informative viewpoints (red cones) from candidate views (gray cones) to minimize reconstruction error between the ground-truth view and the novel view synthesized by rendering methods such as NeRF Mildenhall et al. (2021) or 3DGS Kerbl et al. (2023). **(b) Neural Uncertainty Map (UMap).** For each selected viewpoint, our method predicts a UMap in polar coordinates, assigning uncertainty values to candidate viewpoints on a spherical surface. A viewpoint is parameterized by azimuth $\phi \in [0°, 360°]$ (red) and elevation $\theta \in [0°, 180°]$ (blue), at a fixed radius from the current view. The coordinate system is oriented with the z-axis pointing from the object center to the current view, the y-axis upward, and the x-axis to the right. A sample viewpoint is shown as a black dot, with uncertainty visualized by the colorbar. See **Sec. 2** for details.

takes the current view as input and predicts a neural uncertainty map. This map highlights regions of high ambiguity, enabling selection of informative views based solely on the current observation. UPNet is trained via supervised learning on ground-truth neural uncertainty maps, obtained by comparing single-view 3DGS-synthesized images with ground-truth views using heuristic-based metrics. Compared to prior approaches, our uncertainty maps offer higher interpretability by explicitly visualizing uncertainty across all candidate viewpoints. Since UPNet is lightweight and independent of past viewpoints, it eliminates the need for costly iterative neural rendering model retraining at test time, resulting in 400 times speedup and over 50% reduction in compute usage.

Next, PUN aggregates all the past predicted uncertainty maps, suppresses redundant views with low uncertainty, and selects the next viewpoint with highest uncertainty. Unlike prior methods Mendoza et al. (2020); Wang et al. (2024) that rely on mappings over a fixed, discrete candidate set, PUN generalizes to arbitrary viewpoints.

We evaluate PUN on novel object instances from the same categories as the training set by training a neural rendering model on the selected views and comparing the synthesized outputs to ground truth across pixel-level, semantic, and geometric metrics. PUN achieves reconstruction quality comparable to the all-views upper bound using only half as many views. This performance holds even when using neural rendering models different from those used to generate the neural uncertainty maps. Furthermore, PUN generalizes effectively to AVS for novel object categories and realistic scenes not seen during training. We summarize our key contributions:

**1.** We introduce Peering into the UnkNowN (PUN), a novel AVS method comprising neural uncertainty map prediction and next-best-view selection. Remarkably, PUN achieves reconstruction accuracy comparable to the all-views upper bound using only half as many views.

**2.** We contribute a large-scale Neural Uncertainty Map (NUM) dataset comprising 48 viewpoints and their corresponding neural uncertainty maps across 13 object categories, each with 100 3D object instances. The uncertainty maps are derived using 4 heuristic-based metrics to highlight uncertain regions from single-view view synthesis.

**3.** Compared to competitive baselines, PUN is compute-efficient, highly interpretable, and generalizes well to arbitrary viewpoints and novel object categories. Notably, the viewpoints selected by PUN consistently improve 3D reconstruction performance across different neural rendering models.

## 2 NEURAL UNCERTAINTY MAP (NUM) DATASET

We randomly select 13 object categories, such as sofa, car, and airplane, each with 100 object instances, from the ShapeNet dataset Chang et al. (2015) to construct our Neural Uncertainty Map (NUM) dataset. See **Fig. 2(a)** for the NUM dataset generation pipeline and **Fig. S1** for the list of

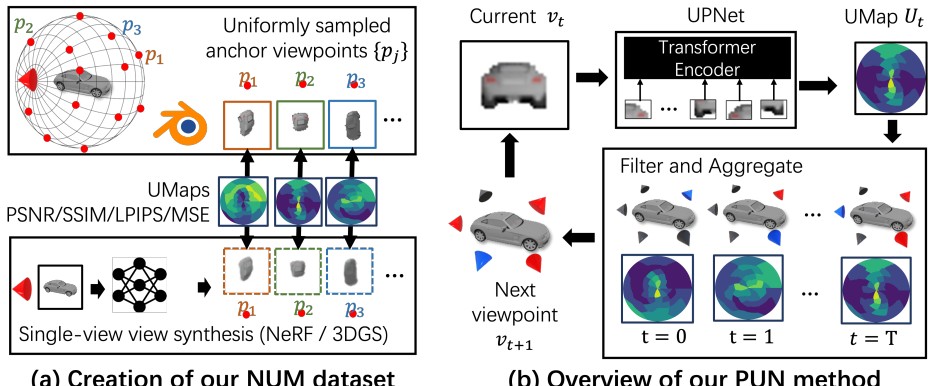

(a) Creation of our NUM dataset          (b) Overview of our PUN method

Figure 2: **(a) Pipeline for Creating the Neural Uncertainty Map (NUM) Dataset.** Given a current viewpoint of a 3D object, its fixed anchor viewpoints $p_j$ (red dots) are defined around a 3D object, with ground-truth images rendered in Blender. A single-view view synthesis method, SplatterImage Szymanowicz et al. (2024) synthesizes novel views at these anchors, and reconstruction errors (PSNR, SSIM, LPIPS, or MSE) are projected into polar coordinates to form the UMaps. **(b) Overview of our proposed Peering into the Unknown (PUN) method for AVS.** From the current view $v_t$, our transformer-based UPNet takes image patches as input and predicts an uncertainty map $U_t$. PUN integrates past UMaps ($t = 0$ to $t$, red triangles), filters redundant viewpoints, and selects the next best view $v_{t+1}$ (blue triangles). This process repeats iteratively to guide exploration.

selected categories. For each object instance, we uniformly sample 48 viewpoints and compute their corresponding neural uncertainty maps (UMaps) as described below. In total, our NUM dataset comprises 62400 pairs of viewpoints and their associated UMaps.

**Neural Uncertainty Maps (UMaps).** Analogous to a depth map estimation in computer vision, given an input viewpoint of a 3D object, one can estimate pixel-wise uncertainty and predict a UMap, as detailed below. In our NUM dataset, exhaustively generating UMaps for all possible input viewpoints is infeasible due to the vast combinatorial space. Therefore, for each 3D object instance, we generate UMaps for 48 selected viewpoints. Specifically, the ShapeNet objects, which are normalized and center-aligned, are directly imported into the Blender simulation engine Community (2018) and the Hierarchical Equal Area isoLatitude Pixelization (HEALPix) algorithm Gorski et al. (2005) with resolution parameter $n_{\text{side}} = 2$ is used to uniformly sample 48 viewpoints on a spherical surface centered at the object's origin, starting from the coordinate (0, 0, 2.73) meters in Blender.

Each of these viewpoints is used to train a neural rendering model for single-view view synthesis. The model can then synthesize novel views from unseen camera poses. We define each camera pose by a fixed radius $r = 2.73$ meters, an elevation angle $\theta \in [0°, 180°]$, and an azimuth angle $\phi \in [0°, 360°]$. The camera is always oriented toward the object center, and $r = 2.73$ meters ensures the object is fully visible but not intersected in Blender. To evaluate the quality of synthesized views, we reuse the same HEALPix method above to generate 48 camera poses on a spherical surface centered at the object's origin, starting from the location of the input viewpoint. In other words, the 48 camera poses for view synthesis are transformed depending on the pose of the input viewpoint, such that their relative positions on the spherical surface remain constant across all input viewpoints.

We name these 48 camera poses as our anchor set $P$ and render ground-truth views in Blender. For each anchor camera pose $p_j \in P$ where $j \in \{1, 2, ..., 48\}$, we also synthesize a corresponding view using the trained neural rendering model. The reconstruction error between the ground-truth and synthesized image is computed using various uncertainty metrics (described below), and this value becomes the uncertainty at anchor point $p_j$ in the UMap. Finally, these anchor points are mapped to polar coordinates to form the UMap. See **Fig. 1(b)** for an illustration of the camera pose setup.

**Single-view view synthesis backbones.** In single-view view synthesis, neural rendering models can be broadly categorized into four types: NeRF-based methods Yu et al. (2021); Wang et al. (2021); Trevithick & Yang (2021), diffusion-based approaches Anciukevičius et al. (2023); Liu et al. (2023); Watson et al. (2022), end-to-end Transformer-based models that take a current view as input and synthesize novel views Hong et al. (2023); Tochilkin et al. (2024), and 3D Gaussian Splatting

(3DGS)-based techniques Szymanowicz et al. (2024); Charatan et al. (2024); Chen et al. (2024). In principle, any of these models could be used as a backbone to synthesize views and compute UMaps. However, due to the high computational cost of view synthesis, training a separate neural rendering model for each of the 48 viewpoints per object instance is impractical. To address this, we adopt Splatter-Image Szymanowicz et al. (2024), pretrained on ShapeNet Chang et al. (2015), as our synthesis backbone for its high compute efficiency and accurate 3D object reconstruction. Built on 3D Gaussian Splatting, Splatter-Image uses a U-Net Ronneberger et al. (2015) to map an input image into one 3D Gaussian per pixel. In ablation studies (see **Sec. 5.1**), we demonstrate that our PUN method, which is trained on UMaps generated by Splatter-Image, generalizes well and can effectively select next-best viewpoints for other neural rendering models.

**Uncertainty measures.** We use four uncertainty measures to quantify differences between synthesized and ground truth views: PSNR, SSIM, LPIPS, and MSE. PSNR (Peak Signal-to-Noise Ratio) measures pixel-wise fidelity and penalizes large intensity deviations. SSIM (Structural Similarity Index) captures structural similarity by accounting for luminance, contrast, and texture. LPIPS (Learned Perceptual Image Patch Similarity) is a perceptual metric based on deep feature representations from pretrained networks Krizhevsky et al. (2017), reflecting semantic similarity. MSE (Mean Squared Error) computes average squared pixel differences and emphasizes raw intensity errors, though it may not align well with human perception.

**Training, validation, and test data split.** To train UPNet, we use data from 11 object categories in the NUM dataset, with 100 instances per category. Each category is divided into training, validation, and test sets with an 8:1:1 split by instance. In addition, 2 extra object categories from the NUM dataset are held out entirely from training and used solely as test data for evaluating UPNet.

## 3 OUR PEERING INTO THE UNKNOWN (PUN) METHOD

An overview of our PUN method is shown in **Fig. 2(b)**. At each timestep $t \in \{0, 1, \ldots, T\}$, given an input view $I_t$ from the current viewpoint $v_t$, PUN first predicts a UMap $U_t$ using a feedforward deep neural network, named the Uncertainty Map Prediction Network (UPNet). We then randomly sample a set of 512 candidate viewpoints $C_t^i$, where $i \in \{1, 2, \ldots, 512\}$. Similar to the anchored camera poses $P_t$, each candidate viewpoint lies on a spherical surface and is parameterized by a fixed radius $r = 2.73$ meters, an elevation angle $\theta \in [0°, 180°]$, and an azimuth angle $\phi \in [0°, 360°]$. The aggregated uncertainty values for these candidates $C_t^i$ are computed based on the sequence of UMaps observed so far, $\{U_1, U_2, \ldots, U_t\}$. PUN then filters out candidate viewpoints with consistently low uncertainty and selects the one with the highest remaining uncertainty as the next best viewpoint.

### 3.1 UNCERTAINTY MAP PREDICTION NETWORK (UPNET)

Given the input view $I_t$ at the current viewpoint $v_t$, the UPNet of PUN predicts a UMap $U_t \in \mathbb{R}^{1 \times 48}$, which is a 48-dimensional vector representing uncertainty values corresponding to the 48 predefined camera poses in the anchor set $P_t$ (see **Sec. 2** for the definition of UMaps in our NUM dataset). To train UPNet, we fine-tune all weights of a Vision Transformer (ViT) Dosovitskiy et al. (2020) pre-trained on ImageNet Deng et al. (2009) for image classification. A fully connected layer is appended to the classification token output of the ViT to generate the predicted UMap $U_t$. The network is supervised using a mean squared error (MSE) loss between the predicted and ground-truth UMaps. In our NUM dataset, UMaps are defined using four different uncertainty measures (**Sec. 2**). For PUN, we use PSNR as the uncertainty measure to construct the ground-truth UMaps. An ablation study in **Sec. 5.3** further investigates the impact of alternative uncertainty measures. See **Sec.C** for additional training and implementation details.

### 3.2 UNCERTAINTY VALUE INTERPOLATION AT THE CANDIDATE VIEWPOINTS

Given the uncertainty values of a UMap $U_t$ at a set of 48 anchor points $P_t$ on a spherical surface, we apply the following interpolation method to estimate the uncertainty at a candidate viewpoint $C_t^i$. For simplicity, we omit the subscript $t$ in the following paragraph.

First, we define the set of neighboring anchor points $\tilde{P} \subset P$ as those whose angular distance to $C^i$ is within 30 degrees. The uncertainty value at $C^i$ is then computed as a weighted sum of the uncertainty values at the neighboring anchors $\tilde{P}$, where the weights are determined by applying a softmax function to the negative angular distances, thus assigning higher weights to closer anchors:

$U^{C_i} = \sum_{\tilde{P}_j \in \tilde{P}} \omega_j U^{\tilde{P}_j}$ and $\omega_j = \frac{e^{-\theta_{ij}}}{\sum_{\tilde{P}_j \in \tilde{P}_i} e^{-\theta_{ij}}}$, where $U^{\tilde{P}_j}$ refers to the uncertainty value at the neighboring anchor point $\tilde{P}_j$ on the UMap, and $\theta_{ij}$ is the angular distance between $\tilde{P}_j$ and $C_i$.

For all past and current UMaps up to timestep $t$, we apply the interpolation method described above to estimate the uncertainty values at each candidate viewpoint $C^i$. For example, we denote by $U_{t-1}^{C^i}$ the interpolated uncertainty from the UMap $U_{t-1}$ at candidate viewpoint $C^i$.

### 3.3 Next Viewpoint Selection

Given the uncertainty values up to timestep $t$ at each candidate viewpoint, we exclude redundant viewpoints whose uncertainty value falls below a threshold of 0.1 at any previous timestep up to $t$. This design is motivated by the observation that candidate viewpoints near previously selected viewpoints are more likely to have already been observed, resulting in lower uncertainty values; thus, they need not be explored again. See the ablation study in **Sec. 5.3** for the effect of excluding redundant viewpoints.

Next, for the remaining candidate viewpoints, since each UMap only reflects the information from a single input image at timestep $t$, we aggregate the uncertainty values at each candidate viewpoint across timesteps 1 to $t$ by multiplying ($\prod$) all the interpolated uncertainty values at $C^i$: $v_{t+1} = \arg\max_{C_i} \prod_{1,2,\ldots,t} U_t^{C^i}$. The candidate with the highest accumulated uncertainty is then selected as the next viewpoint $v_{t+1}$. In **Sec. 5.3**, we also explore alternative aggregation strategies.

## 4 Experiments

For fair comparisons, we evaluate all AVS methods using the same experimental configurations. Each AVS method is allowed to select up to 20 viewpoints. These selected viewpoints are passed to Blender Community (2018) to render 20 corresponding views at a resolution of $512 \times 512$, which are then fed into the same backbone for 3D reconstruction. To avoid any confounding factors during evaluation, all 3D reconstruction backbones are trained from scratch. At each timestep, 512 candidate viewpoints are randomly sampled on a spherical surface of radius 2.73, and all the methods select a viewpoint from this sampled set.

**Datasets.** We evaluate AVS performance across six datasets using object instances from **NUM**, **NeRFAssets** Mildenhall et al. (2021), and **MIP360** Barron et al. (2022). **NUM-inst** comprises 5 unseen instances per category from the same 11 categories used to train UPNet in PUN, while **NUM-cat** contains 10 instances each from 2 novel categories not seen during training. **NUM-light** evaluates the same instances under varied lighting, retaining original illumination and adding a 1000W white point light at (2, –2, 3) m in Blender. **NUM-cam-dist** reduces the camera radius (r = 2.73 / 1.5 m) for **NUM-cat** instances, bringing objects closer and enlarging their projection (see **Fig. S2**). From **NeRFAssets**, we select 4 of 8 instances (Lego, Drums, Hotdog, Materials) with complex geometry and occlusions. **MIP360** contains 9 real-world scenes; we select 4 (Bicycle, Bonsai, Garden, Stump) and retain only views with the target object centered, yielding 97, 146, 93, and 63 views, respectively.

**Baselines.** We include four competitive AVS baselines: **WD** Lee et al. (2022) estimates volumetric uncertainty by computing the entropy of the weight distribution of sampled points along each ray cast from the candidate viewpoint. **A-NeRF** Pan et al. (2022) models NeRF's color outputs as Gaussian distributions and estimates uncertainty based on their variances. **NVF** Xue et al. (2024) incorporates visibility into uncertainty estimation, assigning higher uncertainty to regions that are invisible in the training views. **Uniform** employs Fibonacci sphere sampling to generate a fixed set of uniformly distributed candidate viewpoints over the viewing sphere. **Upper Bound (Upper-bnd)** trains the neural rendering models directly on the ground-truth rendered views corresponding to the evaluation set of 40 camera poses. These same 40 views are then used for evaluating novel view synthesis. Since the training and testing views are identical in this setting, the resulting performance represents an idealized upper bound for novel view synthesis quality.

**Evaluation of 3D reconstruction quality.** We evaluate all AVS methods using the standard neural rendering model **NeRF** Xue et al. (2024), trained for 2,000 iterations on the 20 views selected by each method. In **Sec. 5.1**, we further vary the reconstruction backbone on NUM-3DGS to test whether AVS methods generalize across different backbones. For novel view synthesis evaluation on all datasets except the MIP360 dataset, we randomly sample 40 camera poses uniformly distributed

on the spherical surface surrounding the object. The same set of 40 poses is used for all test object instances and AVS methods to ensure consistent evaluation. For the **MIP360** dataset, where 3D object instances are unavailable, we follow the protocol of Barron et al. (2022) and evaluate using every 8th image in each object's camera view sequence.

Given a test object instance, a 3D reconstruction backbone trained on all the views selected by each AVS method synthesizes novel views. Following prior work Xue et al. (2024), we evaluate the quality of these novel views using three sets of metrics from a fixed set of evaluation camera poses above or a fixed set of evaluation poses from every 8th view in all available views for the MIP360 dataset: **Image Quality.** We report Peak Signal-to-Noise Ratio (PSNR), Structural Similarity Index Measure (SSIM), Learned Perceptual Image Patch Similarity (LPIPS), and Mean Squared Error (MSE) , which respectively measure image fidelity, structural similarity, perceptual differences, and pixel-wise reconstruction error. See **Sec. 2** for more details. **Mesh Quality.** We evaluate accuracy (Acc) and completion ratio (CR) as proposed in Sucar et al. (2021) by comparing the reconstructed mesh—obtained from high-opacity points sampled from the trained NeRF Xue et al. (2024)—against the ground-truth mesh from ShapeNet Chang et al. (2015). Specifically, Acc (cm) is the average distance from points on the reconstructed mesh to their nearest neighbors on the ground-truth mesh, while CR denotes the percentage of points on the reconstructed mesh whose nearest ground-truth points lie within 5 cm. These metrics are not applicable to 3DGS Kerbl et al. (2023) as it does not explicitly produce mesh representations. **Visual Coverage.** We compute Visibility (Vis) and Visible Area (Vis. A.) for evaluation on visibility coverage. See **Sec.C** for the detailed definition of all evaluation metrics.

# 5 RESULTS

## 5.1 OUR PUN ACHIEVES MOST ACCURATE 3D RECON. WITH HIGH COMPUTE EFFICIENCY

**Our PUN outperforms all baselines and matches the upper bound. Tab. 1(a)** presents the 3D reconstruction results on NUM-inst. Our PUN consistently outperforms all baselines across all eight metrics, including image quality, mesh quality, and visual coverage. Notably, even NVF, which explicitly models visual visibility, performs worse than PUN. Although PUN is trained using UMaps derived solely from PSNR, it generalizes well beyond image fidelity, achieving strong performance on geometry-aware and visual coverage metrics. Remarkably, PUN performs on par with the Upper-bnd, despite using only half the number of selected viewpoints.

**Our PUN generalizes to novel object categories.** To assess generalization, we evaluate all AVS methods on **NUM-cat**, as shown in **Tab. 1(b)**. Consistent with the results on **NUM-inst**, our PUN achieves the best performance across all eight metrics, surpassing all baselines. As reported in **Tab. S1**, **Uniform** attains strong visual coverage owing to its even viewpoint distribution, but its performance on other metrics lags behind PUN. Notably, PUN matches **Upper-bound** while requiring only half the training views. Reconstruction quality of all AVS methods as a function of the number of input viewpoints is shown in **Fig. S3**. PUN consistently surpasses all baselines regardless of the number of input viewpoints in most cases.

**The viewpoints selected by PUN are agnostic to the choice of reconstruction backbones.** To test generalization across reconstruction backbones, we introduce the **NUM-3DGS-recon** dataset, which uses the same instances as **NUM-cat** but evaluates AVS methods with the 3D reconstruction backbone **3DGS** Tancik et al. (2023). Using the same selected views and default parameters for all AVS methods, **3DGS** is trained for 30,000 iterations. As shown in **Tab. 1(c)**, PUN consistently outperforms all baselines across all metrics. In addition, we compare our method with the SOTA baseline NVF using Binocular3DGS Han et al. (2024), a few-shot reconstruction method, as the reconstruction backbone. As shown in **Tab. S2**, our method achieves the best performance. Together with the **NUM-cat** results (see **Tab. 1(b)**), these findings confirm that the viewpoints selected by PUN are broadly informative and remain agnostic to the reconstruction backbone.

**PUN generalizes to various lighting conditions, camera distances, and realistic scenes with complex geometry and backgrounds.** PUN demonstrates strong robustness and generalization in 3D reconstruction. Without retraining or fine-tuning, it outperforms baselines on **NUM-light** and **NUM-cam-dist** (**Tab. 1(d)**), showing robustness to changes in lighting and camera distance. It also achieves the best performance on realistic scenes from **NeRFAssets** and **MIP360** (**Tab. 2**), highlighting its ability to select informative views in complex environments.

Table 1: **Evaluation on the AVS performance.** We evaluate the performance of all AVS methods on NUM-inst, NUM-cat, NUM-3DGS-recon, NUM-light and NUM-cam-dist. s1 and s2 refer to experiments on NUM-light and NUM-cam-dist. Average results over 3 runs are reported. See **Tab. S1** for the full results. Best is in **bold** and second best is underlined. For readability, MSE values are scaled by $10^3$ on NUM-3DGS-recon and NUM-cam-dist, and by $10^4$ in other datasets.

| | Method | PSNR↑ | SSIM↑ | LPIPS↓ | MSE↓ | | Method | PSNR↑ | SSIM↑ | LPIPS↓ | MSE↓ |
|---|---|---|---|---|---|---|---|---|---|---|---|
| (a)novel instance | A-NeRF | 32.71 | 0.982 | 0.031 | 8.19 | (b)novel category | A-NeRF | 33.16 | 0.984 | 0.024 | 7.04 |
| | NVF | 33.08 | 0.984 | 0.028 | 6.98 | | NVF | 33.15 | 0.985 | 0.021 | 6.65 |
| | Ours | 33.19 | 0.984 | 0.025 | 6.96 | | Ours | 34.74 | 0.985 | 0.019 | 5.03 |
| | Upper-bnd | **36.47** | **0.989** | **0.017** | **4.11** | | Upper-bnd | **36.91** | **0.990** | **0.013** | **3.33** |
| (c)3DGS recon. | WD | 20.59 | 0.906 | 0.21 | 15.59 | (d)env. setting | NVF-s1 | 31.57 | 0.985 | 0.022 | 11.64 |
| | A-NeRF | 26.22 | 0.948 | 0.12 | 7.50 | | Ours-s1 | **32.84** | **0.987** | **0.018** | **8.32** |
| | NVF | 30.67 | 0.977 | 0.07 | 2.28 | | NVF-s2 | 27.34 | 0.961 | 0.050 | 3.38 |
| | Ours | **36.71** | **0.990** | **0.03** | **0.40** | | Ours-s2 | **31.19** | **0.969** | **0.042** | **1.39** |

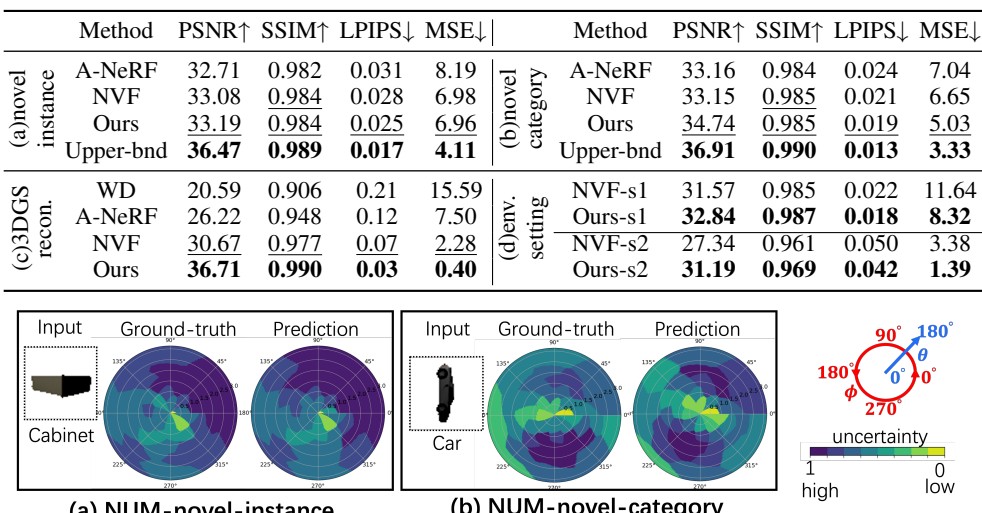

(a) NUM-novel-instance  (b) NUM-novel-category

Figure 3: **Visualization of ground-truth and predicted uncertainty maps by our PUN method.** We present two examples from: (a) NUM-inst and (b) NUM-cat. In each case, the input view $I_t$ at the current viewpoint $v_t$ is fed into UPNet to produce the predicted UMap. The corresponding ground-truth UMap is shown alongside for comparison. Both maps are min-max normalized to $[0, 1]$ for visualization only. See the colorbar for uncertainty values and polar coordinates for the UMaps.

**Our PUN is robust to training on UMaps generated by different novel view synthesis backbones.** We use Splatter-Image Szymanowicz et al. (2024) as the default synthesis backbone for generating UMaps, which are then used to train UPNet in PUN. To test sensitivity to alternative backbones, we replace Splatter-Image with NVF Xue et al. (2024). Since NeRF synthesis is computationally intensive, we train UPNet in PUN on a subset of 50 airplane instances from the NUM dataset, denoting this variant as PUN-NeRF and the corresponding dataset as NeRF-NUM. We then evaluate PUN-NeRF and other AVS methods on 5 airplane instances, using 3DGS as the reconstruction backbone. As shown in **Tab. S3**, PUN-NeRF achieves superior performance on the NeRF-NUM dataset. These results confirm that our method remains robust for AVS, even when trained on UMaps generated by a different synthesis backbone.

**Our PUN predicts reasonable neural uncertainty maps.** To quantitatively evaluate the UMap predictions of our UPNet in PUN, we present the scatter plot of the predicted uncertainty and the ground-truth uncertainty in **Fig. S4**. We compute the Pearson correlation between the ground-truth and predicted uncertainty maps on the test set. The strong correlation coefficient of 0.82 indicates that PUN can reliably predict accurate UMaps and approximate the underlying reconstruction-error landscape. Next, we present two visualization examples of predicted uncertainty maps by PUN on both NUM-inst and NUM-cat (one instance from each dataset) in **Fig. 3**. In **(a)**, given a side view of a cabinet as input, the predicted uncertainty increases with elevation angle $\theta$, represented radially in the polar coordinate. One would expect the uncertainty to also increase along the azimuth angle $\phi$ as the camera rotates $180°$ counter-clockwise toward the cabinet's opposite side. Surprisingly, the uncertainty remains low in these regions likely due to the cabinet's structural symmetry. UPNet may leverage this symmetry to confidently infer the appearance of the unseen side from the visible one. In **(b)**, given a side view of a car, we observe a similar trend in the predicted uncertainty along $\theta$ and $\phi$ as seen on NUM-inst. Notably, the UMap retains a degree of symmetry despite the object class being unseen during training. This consistency may stem from the inherent symmetry bias present in many household objects within the training dataset, which appears to generalize to novel object categories.

**PUN is highly compute-efficient in next-view selection.** Unlike competitive AVS baselines that retrain neural rendering models after each new view—incurring heavy overhead and slow iterative

**More realistic and complex**

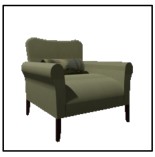 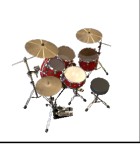 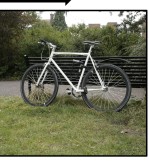

ShapeNet    NeRFAssets    MIP360

Figure 4: **Visualization of the datasets used for AVS evaluation.** From left to right, the object instances exhibit increasing geometric complexity, occlusion, and background clutter.

| Dataset | Method | PSNR↑ | SSIM↑ | LPIPS↓ | MSE↓ |
|---|---|---|---|---|---|
| **NeRFAssets** | NVF | 26.31 | 0.928 | 0.115 | 0.005 |
| | Ours | **26.73** | **0.944** | **0.093** | **0.003** |
| **MIP360** | NVF | 15.41 | 0.203 | 0.653 | 32.03 |
| | Ours | **17.49** | **0.294** | **0.545** | **19.98** |

Table 2: **Evaluation on NeRFAssets and MIP360.** We report the average results over four instances from each of the two datasets. We refer the readers to **Tab. S5** for the full results and the detailed caption. Best is in **bold**.

Table 3: **Ablation Analysis of Key Components in Our PUN Method.** From left to right, we analyze: (a) different uncertainty metrics used to generate ground truth UMaps and train UPNet, (b) the effect of different next-viewpoint selection policies as illustrated in **Sec. 5.3**, and (c) the trade-off between instance diversity and viewpoint density in training samples. In (c), instance diversity refers to the number of object instances per category, while viewpoint density refers to the number of viewpoint–UMap pairs per instance. Best is in **bold** and second best is underlined. MSE values are scaled by $10^4$. See **Tab. S6** for full results in all metrics.

| | (a) Uncertainty metrics | | | | (b) Viewpoint selection policy | | | | | | | (c) Diversity | | | (d) Anchor number | | |
|---|---|---|---|---|---|---|---|---|---|---|---|---|---|---|---|---|---|
| | PSNR (ours) | SSIM | LPIPS | MSE | small+all (ours) | disable+all | top-32+all | single+all | small+last | small+diff | small+add | 80/12 | 40/24 | 20/48 | 12 | 48 | 108 |
| PSNR | **37.4** | 36.4 | 36.1 | 35.0 | **37.4** | 36.9 | 36.8 | 36.9 | 36.9 | 36.3 | 35.7 | **37.0** | 36.8 | 36.8 | 35.4 | 36.0 | **36.1** |

training—PUN uses UPNet, pretrained on our NUM dataset, to predict UMaps directly, avoiding on-the-fly retraining during inference. We benchmark all methods on identical hardware (4× NVIDIA RTX 3090 GPUs, 2× Intel(R) Xeon(R) Silver 4210 CPUs). In **Tab. S4**, PUN achieves up to 400× faster viewpoint selection while drastically reducing resource use: CPU (903% to 74%), RAM (4292 MB to 1870 MB), GPU utilization (30.6% to 0.3%), and GPU memory (8098 MB to 655 MB). Even with the additional 5 minutes for reconstruction after selection, PUN cuts the total runtime from 175 to just 5.5 minutes.

## 5.2 UPNET CAPTURES VIEWPOINT COMPLEXITY

To better understand why UPNet can generalize to previously unseen object categories and remain effective when coupled with different reconstruction backbones, we analyze what cues are implicitly captured by its learned representation. In particular, we investigate whether UPNet's reconstruction-error–based uncertainty measures correlate with basic geometric and texture complexity characteristics of the observed views, which are known to influence the difficulty of single-view reconstruction.

For each anchor viewpoint, we compute five standard view-complexity measures derived from the depth map and the RGB image: (1) depth gradient variance, (2) edge density, (3) color gradient, (4) color entropy, and (5) Laplacian energy. The first two are geometric complexity measures computed from the depth map, while the remaining three are appearance complexity measures computed from the RGB image. See **Sec. E** for detailed descriptions of these measures.

As shown in **Tab. S7**, all metrics show moderate correlation, indicating that viewpoints with greater geometric variance, higher edge density, or richer texture tend to exhibit higher reconstruction error for the single-view synthesis backbone. We also visualize the scatter plot between the complexity metrics and reconstruction error metrics in **Fig. S5** to provide a better understanding of this correlation. By learning such patterns over a large number of diverse objects in ShapeNet, UPNet acquires a generic notion of viewpoint difficulty that is largely domain-agnostic.

## 5.3 ABLATIONS ON OUR PUN REVEAL CRITICAL DESIGN INSIGHTS

**Uncertainty metric ablation.** In NUM, we define four uncertainty metrics—PSNR, SSIM, LPIPS, and MSE—to quantify the difference between synthesized and ground truth views. By default, PUN is trained using ground truth UMaps based on PSNR. To evaluate the effect of different uncertainty measures, we conducted an ablation study by training UPNet with UMaps computed from each

metric. From Tab. 3 (a), the choice of uncertainty metric has limited impact on 3D reconstruction performance using NeRF at inference.

**Ablations on next viewpoint selection.** In PUN, we adopt two key policies: removing redundant viewpoints with low uncertainty (**small**) and aggregating uncertainty across all past and current timesteps (**all**). To evaluate their impact, we explore three variations for redundancy removal and two for uncertainty aggregation. Disabling redundancy removal entirely (**disable**) leads to a noticeable PSNR drop in 3D reconstruction (**Tab. 3(b)**), highlighting the importance of filtering redundant viewpoints. We also test two alternative strategies: (1) excluding candidates ranked among the 32 lowest uncertainty values across previous timesteps (**top-32**), and (2) removing candidates within a $5°$ angular distance of previously selected viewpoints (**single**). Both underperform compared to the default **small + all** in PUN. For uncertainty aggregation, we compare **last**, which selects candidates with the highest uncertainty in the current UMap $U_t$, and **diff**, which favors candidates with the largest uncertainty difference relative to $5°$ neighbors in $U_t$. Both ignore prior information and yield lower performance. Our default **all** aggregation achieves the best results by leveraging temporal consistency across timesteps.

**Ablation on the dataset size.** We perform an ablation study to examine the trade-off between instance diversity and viewpoint density when training UPNet in PUN. To control for total training data, we keep the product of the number of categories, instances per category, and viewpoints per instance constant. As shown in **Tab. 3 (c)**, increasing instance diversity yields greater performance gains than increasing viewpoint density. This suggests that sampling more views from a single object often introduces redundant information, especially in symmetric objects, whereas incorporating a wider range of object instances offers more valuable data variation.

**Ablation on the anchor number.** We further investigate how the number of anchor viewpoints affects UMap quality and downstream view-selection performance. Increasing the anchor count improves angular coverage but also significantly raises the computational cost of NUM generation. As shown in **Tab. 3 (d)**, increasing the anchor number from 12 to 48 provides a noticeable improvement across most metrics, indicating that denser angular sampling leads to more accurate UMaps. However, further increasing to 108 anchors brings only marginal gains, suggesting diminishing returns. Overall, 48 anchors provide a good balance between reconstruction performance and dataset-generation cost, and we adopt this setting as our default configuration.

**Ablation on the redundancy threshold.** The redundancy threshold determines how many candidate viewpoints are removed due to redundancy. A large threshold may discard too many viewpoints, while a small threshold may preserve excessive redundant viewpoints. We evaluate thresholds from 0.0 to 0.3, and the results in **Tab. S6 (e)** show that a threshold of 0.1 provides the best overall performance. Notably, the final reconstruction quality is not highly sensitive to this threshold.

# 6 DISCUSSION

We propose Peering into the UnkNowN (PUN), an efficient AVS method guided by neural uncertainty maps predicted by a lightweight feedforward network, UPNet. Unlike traditional approaches that derive uncertainty from radiance fields, UPNet directly maps a single input image to an uncertainty map over candidate viewpoints. By aggregating these maps over time and applying selection heuristics, PUN identifies informative views with minimal redundancy. Using only half the views of the upper bound, it achieves comparable 3D reconstruction quality while offering up to 400× speedup and 50% resource savings over strong AVS baselines. PUN also generalizes to unseen object categories, varied lighting, camera distances, and realistic scenes, and supports diverse neural rendering models without re-training. Despite its effectiveness, PUN relies on image reconstruction metrics, which may not fully capture geometric fidelity. Incorporating geometry-aware metrics (e.g., mesh quality, visual coverage) into uncertainty prediction could further improve performance. Moreover, PUN currently assumes a spherical viewpoint distribution around a single object; Future work will focus on extending UPNet to handle arbitrary viewpoint distributions in unconstrained 3D environments, as well as integrating it with single-view reconstruction methods to enable online reconstruction and more adaptive viewpoint planning.

## ACKNOWLEDGMENTS

This research is supported by the National Research Foundation, Singapore under its NRFF award NRF-NRFF15-2023-0001 and Mengmi Zhang's Startup Grant from Nanyang Technological University, Singapore.

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

## A  THE USE OF LARGE LANGUAGE MODELS (LLMS)

Large Language Models were used to assist in polishing the writing of this manuscript.

## B  REPRODUCIBILITY STATEMENT

All code, models, and datasets are available at `https://github.com/ZhangLab-DeepNeuroCogLab/PUN`.

## C  ADDITIONAL TRAINING AND IMPLEMENTATION DETAILS

**Training details.** UPNet is trained using the AdamW optimizer Loshchilov & Hutter (2017) with a learning rate of $1 \times 10^{-4}$, a batch size of 32, for 100 epochs on a single NVIDIA RTX A6000 GPU. The original input images of size $248 \times 248$ are center-cropped to $224 \times 224$ and normalized using a mean and standard deviation of 0.5 per channel. We conduct three runs per experiment with different random seeds, following the data splits introduced in **Sec. 2**.

**Evaluation Metrics.** Given a test object instance, a 3D reconstruction backbone trained on all the views selected by each AVS method synthesizes novel views. Following prior work Xue et al. (2024), we evaluate the quality of these novel views using three sets of metrics: **Image Quality.** We report Peak Signal-to-Noise Ratio (PSNR), Structural Similarity Index Measure (SSIM), Learned Perceptual Image Patch Similarity (LPIPS), and Mean Squared Error (MSE), which respectively measure image fidelity, structural similarity, perceptual differences, and pixel-wise reconstruction error. See **Sec. 2** for more details. **Mesh Quality.** We evaluate accuracy (Acc) and completion ratio (CR) as proposed in Sucar et al. (2021) by comparing the reconstructed mesh—obtained from high-opacity points sampled from the trained NeRF Xue et al. (2024)—against the ground-truth mesh from ShapeNet Chang et al. (2015). Specifically, Acc (cm) is the average distance from points on the reconstructed mesh to their nearest neighbors on the ground-truth mesh, while CR denotes the percentage of points on the reconstructed mesh whose nearest ground-truth points lie within 5 cm. These metrics are not applicable to 3DGS Kerbl et al. (2023) as it does not explicitly produce mesh representations. **Visual Coverage.** We compute Visibility (Vis) as the proportion of ground-truth mesh faces that are directly visible by intersecting a ray from at least one selected viewpoint without occlusion. Visible Area (Vis. A.) measures the total surface area of these visible faces relative to the entire surface area of the ground-truth mesh. Notably, both metrics are independent of the neural rendering models and solely depend on the selected viewpoints.

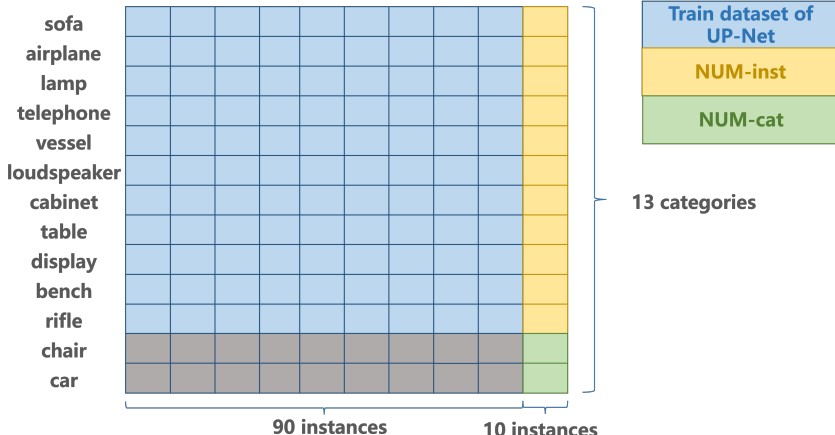

Figure S1: **The data distribution of our NUM dataset.** In total, we select 13 categories from the ShapeNet dataset Chang et al. (2015), with 100 object instances per category used to generate our NUM dataset. The blue regions indicate the training dataset for our UPNet, consisting of 90 instances from each of 11 categories. The yellow regions represent the instances in NUM-inst. The green regions show the instances in NUM-cat.

## D   DETAIL EXPERIMENT RESULTS

Table S1: **Evaluation on the view selection performance of PUN.** We evaluate the performance of PUN on NUM-inst, NUM-cat, NUM-3DGS-recon, NUM-light and NUM-cam-dist. s1 and s2 separately refers to experiments on NUM-light and NUM-cam-dist. Average results over 3 runs are reported, with standard deviations shown in the brackets. For the experiment on NUM-3DGS-recon, since we use 3DGS as the reconstruction backbone, it is not possible to extract a predicted mesh to compute the mesh quality metric, which is therefore denoted by "–". Best is in **bold** and second best is underlined. For readability, MSE values are scaled by $10^3$ in the experiment on NUM-3DGS-recon and NUM-cam-dist, and by $10^4$ in other experiments.

| Evaluation setting | Method | PSNR↑ | SSIM↑ | LPIPS↓ | MSE↓ | Acc.↓ | CR↑ | Vis.↑ | Vis. A.↑ |
|---|---|---|---|---|---|---|---|---|---|
| | | (a) Evaluation on NUM-inst | | | | | | | |
| novel instance | WD | 31.92 (0.34) | 0.979 (0.001) | 0.035 (0.002) | 8.55 (0.71) | 0.36 (0.01) | 0.14 (0.01) | 0.33 (0.010) | 0.72 (0.011) |
| | A-NeRF | 32.71 (0.41) | 0.982 (0.001) | 0.031 (0.003) | 8.19 (0.54) | 0.36 (0.01) | 0.16 (0.01) | 0.37 (0.014) | 0.77 (0.006) |
| | NVF | 33.08 (0.44) | 0.984 (0.001) | 0.028 (0.002) | 6.98 (0.72) | 0.36 (0.01) | **0.18** (0.01) | 0.39 (0.013) | 0.79 (0.009) |
| | Uniform | 31.14 (0.66) | 0.968 (0.002) | 0.049 (0.004) | 14.93 (1.85) | **0.35** (0.01) | 0.17 (0.01) | 0.41 (0.016) | 0.81 (0.014) |
| | Ours | 33.19 (1.74) | 0.984 (0.005) | 0.025 (0.004) | 6.96 (0.24) | **0.35** (0.01) | 0.18 (0.01) | 0.41 (0.018) | 0.82 (0.010) |
| | Upper-bnd | **36.47** (0.67) | **0.989** (0.001) | **0.017** (0.001) | **4.11** (0.05) | 0.35 (0.01) | 0.17 (0.01) | **0.49** (0.018) | **0.85** (0.009) |
| | | (b) Evaluation on NUM-cat | | | | | | | |
| novel category | WD | 31.93 (2.17) | 0.980 (0.006) | 0.029 (0.013) | 8.34 (0.44) | 0.35 (0.09) | 0.14 (0.05) | 0.30 (0.066) | 0.67 (0.023) |
| | A-NeRF | 33.16 (2.28) | 0.984 (0.006) | 0.024 (0.013) | 7.04 (0.32) | 0.35 (0.09) | 0.15 (0.04) | 0.34 (0.088) | 0.72 (0.007) |
| | NVF | 33.15 (2.42) | 0.985 (0.005) | 0.021 (0.010) | 6.65 (0.40) | 0.35 (0.09) | **0.19** (0.06) | 0.35 (0.084) | 0.75 (0.024) |
| | Uniform | 31.62 (2.91) | 0.972 (0.010) | 0.037 (0.020) | 11.59 (8.94) | 0.34 (0.07) | 0.18 (0.06) | 0.37 (0.100) | 0.78 (0.040) |
| | Ours | 34.74 (2.81) | 0.985 (0.005) | 0.019 (0.006) | 5.03 (0.36) | 0.34 (0.07) | 0.18 (0.05) | 0.37 (0.097) | 0.78 (0.037) |
| | Upper-bnd | **36.91** (2.56) | **0.990** (0.003) | **0.013** (0.006) | **3.33** (0.21) | 0.34 (0.07) | 0.17 (0.04) | **0.44** (0.111) | **0.81** (0.026) |
| | | (c) Evaluation on NUM-3DGS-recon | | | | | | | |
| 3DGS | WD | 20.59 (1.47) | 0.906 (0.013) | 0.21 (0.03) | 15.59 (1.88) | - | - | - | - |
| | A-NeRF | 26.22 (2.47) | 0.948 (0.018) | 0.12 (0.03) | 7.50 (3.56) | - | - | - | - |
| | NVF | 30.67 (2.11) | 0.977 (0.007) | 0.07 (0.02) | 2.28 (1.63) | - | - | - | - |
| | Ours | **36.71** (1.17) | **0.990** (0.002) | **0.03** (0.01) | **0.40** (0.15) | - | - | - | - |
| | | (d) Evaluation on NUM-light and NUM-cam-dist | | | | | | | |
| Point lighting | NVF | 31.57 (3.13) | 0.985 (0.007) | **0.02** (0.013) | 11.64 (10.08) | 0.34 (0.08) | **0.19** (0.07) | **0.44** (0.173) | **0.80** (0.04) |
| | Ours | **32.84** (3.23) | **0.987** (0.007) | **0.02** (0.010) | **8.32** (7.11) | **0.31** (0.08) | 0.18 (0.07) | 0.43 (0.169) | 0.79 (0.064) |
| Camera distance | NVF | 27.34 (3.47) | 0.961 (0.020) | 0.05 (0.027) | 3.38 (3.35) | 0.38 (0.10) | 0.18 (0.08) | 0.42 (0.094) | 0.77 (0.003) |
| | Ours | **31.19** (3.06) | **0.969** (0.013) | **0.04** (0.023) | **1.39** (1.07) | **0.36** (0.07) | **0.19** (0.05) | **0.43** (0.108) | **0.81** (0.032) |

Table S2: **Evaluation of AVS Methods using Binocular3DGS as the reconstruction backbone.** Best is in **bold**. MSE values are scaled by $10^4$ for readability.

| Method | PSNR↑ | SSIM↑ | LPIPS↓ | MSE↓ |
|--------|-------|-------|--------|------|
| NVF | 41.17 | 0.992 | 0.012 | 12.72 |
| ours | **43.78** | **0.996** | **0.007** | **6.15** |

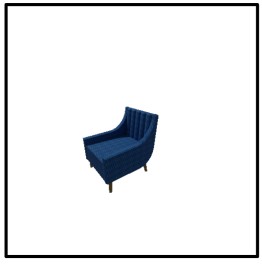 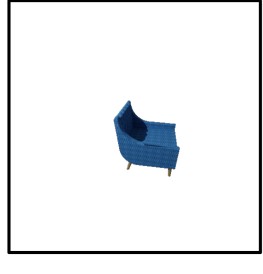 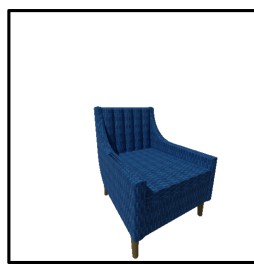

**NUM-cat**   **NUM-light**   **NUM-cam-dist**

Figure S2: **Visualization of NUM-light and NUM-cam-dist**. We select a chair instance to illustrate how different environment settings affect the input view. In NUM-light, the added point light source results in more pronounced shadows caused by occlusion from the chair back. In NUM-cam-dist, the reduced camera distance makes the object appear larger in the input view.

## E  FURTHER DISCUSSION

**UPNet learns a generic notion of viewpoint difficulty.** To better understand why UPNet can generalize to previously unseen object categories and remain effective when coupled with different reconstruction backbones, we analyze what cues are implicitly captured by its learned representation. In particular, we investigate whether UPNet's reconstruction-error–based supervision correlates with basic geometric and texture complexity characteristics of the observed views, which are known to influence the difficulty of single-view reconstruction.

For each anchor viewpoint, we compute five standard view-complexity measures derived from the depth map and RGB image:(1)depth gradient variance — the variance of the depth-gradient magnitude, where the gradient is computed using Sobel filters. This captures geometric relief, surface curvature changes, and depth discontinuities; (2)edge density — the ratio of edge pixels in the depth map, obtained by applying a Canny edge detector. It reflects the amount of geometric structure visible from the viewpoint; (3)color gradient — the mean magnitude of image-space color gradients (Sobel-based), measuring the strength and density of texture variations; (4)color entropy — the entropy of the RGB color histogram, quantifying the diversity and complexity of the appearance distribution; (5)laplacian energy — the average absolute response of the Laplacian operator on the RGB image, capturing high-frequency texture, fine details, and sharp edges.

This also explains that predicting the reconstruction error of a single-view 3DGS model is a valid and effective proxy for guiding multi-view NeRF reconstruction. Reconstruction difficulty is largely determined by the geometry and visibility structure of the object, rather than the specific reconstruction backbone. Viewpoints that are occluded or geometrically complex tend to produce higher errors for any reconstruction model—single-view or multi-view—because all such models ultimately optimize the same objective of minimizing view-dependent reconstruction error as shown in **Tab.S7**. As a result, the error patterns produced by a single-view 3DGS predictor reflect a general notion of "viewpoint difficulty" that is shared across model classes.

By training UPNet to approximate this error landscape across many objects, the network learns to identify viewpoints that are inherently hard or easy to reconstruct, independent of the downstream backbone.

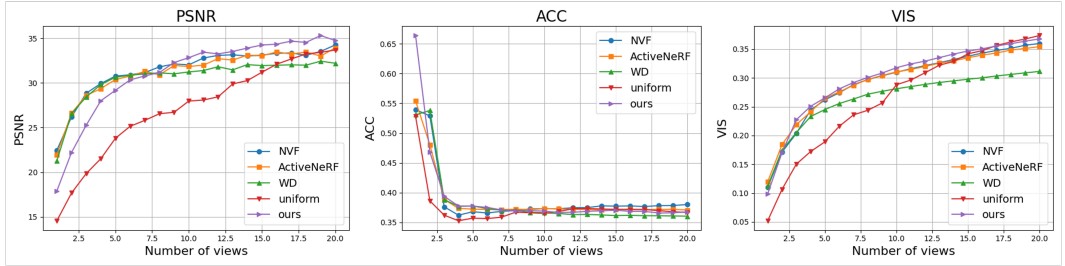

Figure S3: **Visualization of reconstruction performance as a function of the number of views.** We present the trends of PSNR, accuracy, and visibility with respect to the number of selected views on NUM-cat, serving as representative metrics from each of the three category of metrics. The horizontal axis indicates the number of selected views, while the vertical axis shows the corresponding metric values.

Table S3: **Evaluation of AVS Methods on NeRF-NUM.** Best is in **bold** and second best is underlined. MSE values are scaled by $10^3$ for readability.

| Method | PSNR↑ | SSIM↑ | LPIPS↓ | MSE↓ |
|---|---|---|---|---|
| WD | 18.92 | 0.895 | 0.270 | 19.09 |
| A-NeRF | 21.96 | 0.921 | 0.198 | 13.94 |
| NVF | 26.40 | 0.958 | 0.122 | 6.25 |
| PUN-NeRF | **35.69** | **0.991** | **0.049** | **0.51** |

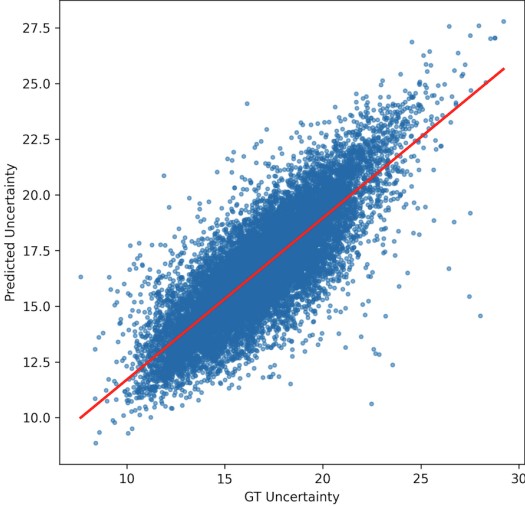

Figure S4: **Scatter plot comparing predicted uncertainty with the ground-truth uncertainty.** Uncertainty points are sampled from the test set. The red line denotes the fitted linear trend.

Table S4: **Processing Time and Resource Comparison.** We report the average time, and computing resources (from left to right: CPU usage, RAM, GPU usage, and GPU memory) required to select 20 viewpoints for each AVS method during inference. Average results over 3 runs are reported, with standard deviations shown in the brackets. Best is in **bold** and the second best is underlined.

| Method | Time (min) | CPU (%) | RAM (MB) | GPU (%) | GPU (MB) |
|---|---|---|---|---|---|
| WD | 198 (19) | 630 (68) | 4354 (52) | 28.52 (1.82) | 8144 (271) |
| A-NeRF | 184 (13) | 678 (104) | 4308 (114) | 29.60 (1.95) | 8043 (97) |
| NVF | 175 (19) | 903 (86) | 4292 (54) | 30.59 (2.09) | 8098 (38) |
| Ours | **0.5** (0.13) | **80** (14) | **1870** (250) | **0.28** (0.06) | **655** (70) |

Table S5: **Evaluation on NeRFAssets and MIP360 dataset.** We report the average results over four instances from each of the two datasets. For the MIP360 dataset, since ground-truth geometry is not available, the mesh quality and visual coverage metrics cannot be computed and are therefore denoted by "–". Best is in **bold**. MSE values are scaled by $10^3$ for the experiment on MIP360 dataset and by $10^4$ for the experiments on NeRFAssets.

| **NeRFAssets** | PSNR↑ | SSIM↑ | LPIPS↓ | MSE↓ | Acc.↓ | CR↑ | Vis.↑ | Vis. A.↑ |
|---|---|---|---|---|---|---|---|---|
| NVF | 26.31 | 0.928 | 0.115 | 0.005 | 0.294 | **0.51** | **0.457** | **0.64** |
| Ours | **26.73** | **0.944** | **0.093** | **0.003** | **0.291** | 0.47 | 0.424 | 0.62 |
| **MIP360** | PSNR↑ | SSIM↑ | LPIPS↓ | MSE↓ | Acc.↓ | CR↑ | Vis.↑ | Vis. A.↑ |
| NVF | 15.41 | 0.203 | 0.653 | 32.03 | - | - | - | - |
| Ours | **17.49** | **0.294** | **0.545** | **19.98** | - | - | - | - |

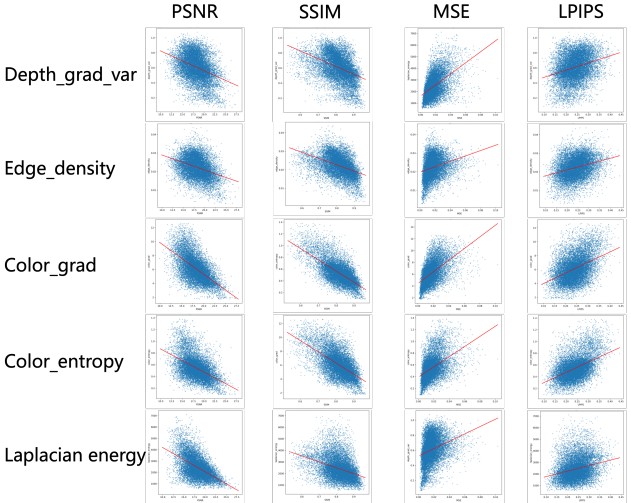

Figure S5: **Scatter plots illustrating the relationship between reconstruction error and view-complexity indicators.** Each subplot shows one reconstruction-error metric (PSNR, SSIM, MSE, or LPIPS) plotted against one of the five complexity measures (depth gradient variance, edge density, color gradient, color entropy, laplacian energy). Blue dots represent sampled viewpoints from the test set, and the red line denotes the fitted linear correlation trend for that metric pair.

Table S6: **Ablation experiment on key components in PUN,** including the different viewpoint selection policies, different uncertainty metrics used for guiding the selection, and instance diversity and viewpoint density in the UPNet Training dataset. Instance diversity refers to the number of instances per category in the training set, while viewpoint density denotes the number of data samples (i.e., input image–UMap pairs) per instance. **Bold** indicates the best performance for each metric, while underline indicates the second best. (MSE values are scaled by $10^4$ for readability.)

| (a) Ablation on different uncertainty metrics | | | | | | | | |
|---|---|---|---|---|---|---|---|---|
| metric | PSNR ↑ | SSIM ↑ | LPIPS ↓ | MSE ↓ | Acc. ↓ | CR ↑ | Vis. ↑ | Vis. A. ↑ |
| PSNR (ours) | **37.41** | **0.990** | **0.015** | **2.50** | 0.37 | 0.16 | **0.37** | **0.74** |
| SSIM | 36.37 | 0.989 | 0.016 | 2.90 | 0.37 | **0.18** | 0.36 | 0.73 |
| LPIPS | 36.09 | 0.989 | **0.015** | 3.23 | 0.38 | 0.17 | 0.35 | 0.72 |
| MSE | 34.95 | 0.870 | 0.017 | 4.11 | **0.36** | 0.14 | 0.33 | 0.68 |
| (b) Ablation on different viewpoint selection policy | | | | | | | | |
| policy | PSNR ↑ | SSIM ↑ | LPIPS ↓ | MSE ↓ | Acc. ↓ | CR ↑ | Vis. ↑ | Vis. A. ↑ |
| small+all (ours) | **37.41** | **0.990** | 0.015 | **2.50** | 0.37 | 0.16 | **0.37** | **0.74** |
| disable+all | 36.85 | 0.989 | 0.014 | 2.38 | 0.37 | 0.16 | **0.37** | 0.73 |
| top-32+all | 36.80 | **0.990** | **0.013** | 3.40 | 0.37 | 0.16 | **0.37** | 0.73 |
| single+all | 36.88 | 0.989 | 0.014 | 2.59 | 0.37 | 0.17 | **0.37** | **0.74** |
| diable+last | 35.82 | 0.989 | 0.016 | 3.44 | 0.36 | 0.16 | 0.36 | 0.70 |
| small+last | 36.94 | **0.990** | 0.014 | 2.51 | 0.36 | 0.16 | 0.36 | 0.72 |
| top-32+last | 37.00 | 0.990 | **0.013** | 2.55 | 0.36 | 0.17 | **0.37** | 0.72 |
| single+last | 36.16 | 0.989 | 0.015 | 3.12 | 0.35 | 0.17 | 0.36 | 0.72 |
| diable+diff | 31.22 | 0.980 | 0.027 | 10.44 | 0.35 | 0.14 | 0.30 | 0.64 |
| small+diff | 36.27 | 0.989 | 0.015 | 3.36 | 0.35 | 0.16 | 0.36 | 0.72 |
| top-32+diff | 33.08 | 0.983 | 0.023 | 7.50 | **0.33** | 0.17 | 0.32 | 0.67 |
| single+diff | 34.71 | 0.987 | 0.017 | 4.00 | 0.35 | 0.17 | 0.34 | 0.71 |
| small+add | 35.68 | 0.985 | 0.021 | 3.74 | 0.37 | 0.15 | 0.36 | 0.72 |
| (c) Ablation on different instance diversity and viewpoint sample density | | | | | | | | |
| diversity/ density | PSNR ↑ | SSIM ↑ | LPIPS ↓ | MSE ↓ | Acc. ↓ | CR ↑ | Vis. ↑ | Vis. A. ↑ |
| 80/12 | **36.96** | **0.990** | **0.014** | 2.97 | **0.37** | 0.17 | **0.37** | 0.73 |
| 40/24 | 36.76 | 0.989 | **0.014** | 2.69 | **0.37** | **0.19** | 0.36 | 0.73 |
| 20/48 | 36.82 | **0.990** | **0.014** | **2.68** | **0.37** | 0.17 | **0.37** | **0.74** |
| (d) Ablation on different anchor numbers | | | | | | | | |
| anchor number | PSNR ↑ | SSIM ↑ | LPIPS ↓ | MSE ↓ | Acc. ↓ | CR ↑ | Vis. ↑ | Vis. A. ↑ |
| 12 | 35.37 | 0.992 | 0.014 | 3.01 | **0.33** | **0.19** | **0.18** | 0.66 |
| 48 | 35.97 | 0.992 | 0.014 | 2.78 | 0.34 | 0.17 | 0.17 | 0.64 |
| 108 | **36.13** | **0.993** | **0.013** | **2.60** | 0.34 | 0.17 | **0.18** | **0.68** |
| (e) Ablation on different redundancy threshold | | | | | | | | |
| redundancy threshold | PSNR ↑ | SSIM ↑ | LPIPS ↓ | MSE ↓ | Acc. ↓ | CR ↑ | Vis. ↑ | Vis. A. ↑ |
| 0.0 | 35.30 | **0.986** | **0.019** | 4.51 | 0.37 | **0.15** | **0.37** | 0.72 |
| 0.1 | **36.04** | **0.986** | 0.020 | 4.75 | 0.37 | **0.15** | 0.36 | **0.73** |
| 0.2 | 35.70 | **0.986** | 0.020 | 5.19 | 0.37 | **0.15** | **0.37** | **0.73** |
| 0.3 | 35.81 | 0.985 | 0.021 | **4.13** | **0.36** | 0.14 | 0.36 | **0.73** |

Table S7: Correlation between basic geometric/texture complexity indicators and reconstruction error metrics. Negative values (↓) indicate that higher complexity correlates with **lower** PSNR/SSIM (worse reconstruction), while positive values (↑) indicate correlation with **higher** MSE/LPIPS (higher error).

| | PSNR | SSIM | MSE | LPIPS |
|---|---|---|---|---|
| depth_grad_var | −0.36 ↓ | −0.40 ↓ | 0.28 ↑ | 0.28 ↑ |
| edge_density | −0.35 ↓ | −0.41 ↓ | 0.26 ↑ | 0.29 ↑ |
| color_grad | −0.60 ↓ | −0.60 ↓ | 0.56 ↑ | 0.43 ↑ |
| color_entropy | −0.50 ↓ | −0.69 ↓ | 0.46 ↑ | 0.48 ↑ |
| laplacian_energy | −0.56 ↓ | −0.38 ↓ | 0.53 ↑ | 0.25 ↑ |

