# OpenReview forum: "PEERING INTO THE UNKNOWN: ACTIVE VIEW SELECTION WITH NEURAL UNCERTAINTY MAPS FOR 3D RECONSTRUCTION"
_ICLR.cc/2026/Conference — ICLR 2026 Poster_

### Official Review · Reviewer_bRcV · 2025-10-19

**Soundness:** 3
**Presentation:** 3
**Contribution:** 3
**Rating:** 6
**Confidence:** 2

**Summary:**

This paper introduces PUN, a novel and highly efficient method for Active View Selection (AVS) in 3D reconstruction. Instead of the conventional, computationally expensive approach of retraining a neural rendering model at each step, PUN employs a lightweight network, UPNet, to directly predict a "Neural Uncertainty Map" from a single input view. This map guides the selection of the next most informative viewpoint. To enable this, the authors created a large-scale dataset (NUM) of images paired with pre-computed uncertainty maps. The method demonstrates significant improvements in speed (up to 400x) and computational efficiency while achieving reconstruction quality comparable to using a full set of views, and impressively generalizes to novel objects and real-world scenes without retraining.

**Strengths:**

- The core contribution is reformulating the AVS problem from an iterative optimization into a direct prediction task. This is a significant conceptual shift that effectively breaks the efficiency bottleneck plaguing prior AVS methods for neural rendering.
- The experimental validation is extensive and convincing. The method is tested across multiple datasets ranging from synthetic objects (ShapeNet) to complex real-world scenes (NeRFAssets, MIP360), demonstrating robust generalization. The ablation studies are thorough and provide clear insights into the method's design.

**Weaknesses:**

- The "ground truth" for the uncertainty maps is generated using the reconstruction error from a single synthesis model (Splatter-Image). UPNet may be learning the specific failure modes of this proxy model rather than a universal notion of 3D uncertainty, potentially limiting its theoretical soundness.
- Several key design choices lack clear justification. For instance, the paper does not provide a strong rationale for using multiplicative aggregation of uncertainty over time, which is highly sensitive to low values, compared to simpler alternatives like addition.
- The paper demonstrates impressive generalization from synthetic to real-world data but offers limited insight into why this works so well. The discussion on learning priors like object symmetry is speculative and would be strengthened by targeted experiments.

**Questions:**

The "ground truth" uncertainty is dependent on the Splatter-Image model. How sensitive is UPNet's performance if the NUM dataset were generated with a different backbone model, for example, a NeRF-based one? Does the model learn general geometric uncertainty or just the artifacts of the proxy?

Could you elaborate on the choice of the aggregation for historical uncertainty values? What are the theoretical or empirical advantages of this strategy over additive aggregation or simply taking the maximum uncertainty?

The paper speculates that generalization stems from learning priors like symmetry. Have you considered experiments on highly asymmetric or topologically complex objects (e.g., a tangled rope) to test the limits of this learned prior?

How were the hyperparameters, such as the 48 anchor points for the UMap and the 0.1 redundancy filter threshold, determined? How sensitive is the final reconstruction performance to these choices?

---

> ### Author Response · Authors · 2025-11-21
> **Response part 1**
>
> **bRcV.1 - Weakness: The "ground truth" for the uncertainty maps is generated using the reconstruction error from a single synthesis model (Splatter-Image). UPNet may be learning the specific failure modes of this proxy model rather than a universal notion of 3D uncertainty, potentially limiting its theoretical soundness.**
>
> response: We appreciate the reviewer’s insightful comment. As discussed in our response to **yuxC.7**, we further examine whether basic geometric and texture–complexity indicators correlate with the reconstruction error signal used in our UMap supervision. For each anchor viewpoint, we compute several standard view-complexity measures: (1) depth_grad_var: variance of depth gradients, capturing geometric relief and surface discontinuities; (2) edge_density: proportion of high-gradient pixels in the depth image, reflecting structural detail; (3) color_grad: magnitude of color gradients, indicating texture richness; (4) color_entropy: entropy of the color histogram, measuring appearance diversity; (5) laplacian_energy: Laplacian magnitude, capturing high-frequency texture and edges. We refer the reviewer to the line 1113 of the revised appendix for a detailed introduction of these complexity metrics.
>
> As shown in **Tab. S7** in the revised appendix, all metrics show moderate correlation, indicating that viewpoints with greater geometric relief, higher edge density, or richer texture tend to exhibit higher reconstruction error for the single-view synthesis backbone. We also visualize the scatter plot between the complexity metrics and reconstruction error metrics in **Fig. S7** to provide a better understanding of this correlation.
>
> By learning such patterns over a large number of diverse objects in ShapeNet, UPNet acquires a generic notion of viewpoint difficulty that is largely domain-agnostic.
> Empirically, this is supported by our experiments on the MIP360 dataset (**Tab. 3**), where UPNet—trained solely on synthetic, object-centric data—still produces meaningful UMaps and guides competitive multi-view reconstruction performance. This indicates that the model does not rely on dataset-specific appearance cues, but instead captures geometry and texture complexities that remain valid in real 360° environments.
>
> Empirically, this shared structure is reflected in our results: UMaps learned from Splatter-Image reliably guide multi-view reconstruction across different backbones and datasets (**Tab.1** **Tab. 3**). This suggests that UPNet is not merely capturing model-specific artifacts but learning a more general notion of viewpoint difficulty. We have included these results and discussion in line 1113 of the revised appendix.
>
> **bRcV.2 - Weakness: Several key design choices lack clear justification. For instance, the paper does not provide a strong rationale for using multiplicative aggregation of uncertainty over time, which is highly sensitive to low values, compared to simpler alternatives like addition.**
>
> response: We thank the reviewer for pointing out this design choice. Our rationale for using multiplicative aggregation is that when a viewpoint receives a low uncertainty value at any step, it implies that the corresponding region has already been sufficiently reconstructed and it should not be revisited. Multiplicative aggregation naturally captures this behavior: a single low value down-weights that viewpoint in subsequent iterations, preventing the algorithm from revisiting regions that have already been resolved. In contrast, with additive aggregation, a viewpoint that once achieved low uncertainty may still be repeatedly selected if its uncertainty happens to be higher at other steps, which contradicts the intended behavior of “avoiding already reconstructed regions.”
>
> To further support this choice, we conducted an ablation study comparing multiplicative and additive aggregation. As shown in **Tab. S9**, multiplicative aggregation achieves consistently better performance across multiple metrics.
>
> We have also evaluated different viewpoint-selection strategies more broadly (**Tab. 4**) and discuss them in line 474 of the revised main text and line 1185 of the appendix, where multiplicative aggregation demonstrates the strongest overall performance.

---

> > ### Author Response · Authors · 2025-11-21
> > **Response part 2**
> >
> > **bRcV.3 - Weakness: The paper demonstrates impressive generalization from synthetic to real-world data but offers limited insight into why this works so well. The discussion on learning priors like object symmetry is speculative and would be strengthened by targeted experiments.**
> >
> > response: We acknowledge the reviewer’s concern. As discussed in our response to**bRcV.1**, UPNet’s ability to generalize stems from the fact that the reconstruction-error landscape is primarily shaped by geometric structure and texture complexity—factors that are largely domain-agnostic. This observation is quantitatively supported by the correlations reported in **Tab. S7** and **Fig. S7** of the revised appendix, which shows that several view-complexity indicators (e.g., depth-gradient variance, edge density, texture gradients) exhibit moderate correlation with reconstruction error. We refer the reviewer to the line 1113 of the revised appendix for detailed introduction of these complexity metrics and discussion.
> >
> > By training on a large number of diverse ShapeNet objects, UPNet learns these structural regularities and consequently acquires a generic notion of viewpoint difficulty that is not tied to any specific dataset. Empirically, this is confirmed by the MIP360 experiments (**Tab. 3**), where a model trained solely on synthetic, object-centric data still produces meaningful UMaps and guides competitive multi-view reconstruction in cluttered, real-world 360° scenes.
> > Taken together, these results demonstrate that reconstruction error provides a domain-agnostic supervisory signal for active view selection, enabling the learned prior to transfer across datasets and reconstruction backbones. We refer the reviewer to our detailed discussion in the response to **bRcV.1** and note that the full results and accompanying analysis is included in line 1113 of the revised appendix.
> >
> > **bRcV.4 - Question: The "ground truth" uncertainty is dependent on the Splatter-Image model. How sensitive is UPNet's performance if the NUM dataset were generated with a different backbone model, for example, a NeRF-based one? Does the model learn general geometric uncertainty or just the artifacts of the proxy?**
> >
> > response: We thank the reviewer for the thoughtful question. We refer the reviewer to our NUM-NeRF experiment (**Tab. S2**), where the ground truth’ UMaps were generated not by Splatter-Image but by a NeRF-based single-view synthesis model. Due to the computational cost of NeRF synthesis, UPNet was trained on a subset of 50 airplane instances (NeRF-NUM) and evaluated on 5 airplane instances using 3DGS as the reconstruction backbone. As shown in **Tab. S2** and discussed in line 405 of the revised main text, PUN-NeRF still achieves the best performance among all AVS baselines, demonstrating that UPNet remains effective even when trained using UMaps from a completely different synthesis backbone.
> >
> > More broadly, consistent with our response to **bRcV.1**, these results indicate that UPNet captures a generic heuristic of viewpoint reconstruction difficulty, largely governed by geometric and visibility structures rather than model-specific failure modes. The fact that UMaps generated from two very different backbones (Splatter-Image and NeRF) both lead to strong view-selection performance suggests that the learned prior reflects general geometric uncertainty, not artifacts of a specific proxy model.
> >
> > We have included these findings and clarifications in the line 1113 of the revised appendix.
> >
> > **bRcV.5 - Question: Could you elaborate on the choice of the aggregation for historical uncertainty values? What are the theoretical or empirical advantages of this strategy over additive aggregation or simply taking the maximum uncertainty?**
> >
> > response: We thank the reviewer for the insightful question. Please refer to the response to **bRcV.2** to see the experiments results on additive aggregation policy and detailed discussion.
> >
> > Beyond pairwise comparison with additive aggregation, we also evaluated a policy that greedily selects the viewpoint with maximum uncertainty at each step, corresponding to a “max-uncertainty” strategy. As shown in **Tab. 3(b)**, **Tab. S5(b)**, and discussed in line 1185 of the revised appendix, this strategy performs significantly worse due to its lack of historical context.
> >
> > Overall, based on the ablation result, the multiplicative aggregation strategy yields the strongest performance.

---

> > > ### Author Response · Authors · 2025-11-21
> > > **Response part 3**
> > >
> > > **bRcV.6 - Question: The paper speculates that generalization stems from learning priors like symmetry. Have you considered experiments on highly asymmetric or topologically complex objects (e.g., a tangled rope) to test the limits of this learned prior?**
> > >
> > > response: We thank the reviewer for this valuable question. As discussed in our response to **bRcV.3**, the generalization ability of UPNet stems from the fact that the reconstruction error landscape is governed primarily by geometric and visibility structures rather than dataset-specific appearance cues. By training on a large number of diverse ShapeNet objects, UPNet learns a generic notion of viewpoint difficulty—for example, views with limited visible area or ambiguous geometry naturally produce higher reconstruction error. These structural patterns arise consistently across both synthetic and real-world environments.
> > >
> > > Our analysis shows that the model behaves as expected on symmetric objects but can fail on asymmetric ones. For symmetric shapes—for example (**Fig. S5**), a table with bilateral symmetry—the single-view model correctly reconstructs the views of the input viewpoint (red) and its symmetric counterpart (green), because both reveal comparable geometric information. In contrast, for asymmetric objects such as a bench with an armrest on only one side, the model sometimes incorrectly assumes symmetry (**Fig. S6**): although the input view reveals the armrest only on one side, the model still predicts the same at the mirrored viewpoint, leading to larger reconstruction error. This failure arises because, under single-view ambiguity, the backbone tends to hallucinate missing geometry by defaulting to symmetry priors.  These observations clarify both the strengths and limitations of UPNet: it captures real symmetries reliably but may misinterpret asymmetries when the input view lacks sufficient cues.
> > >
> > > We have included these visualizations and an expanded discussion in line 1211 of the revised appendix to clarify these insights
> > >
> > > **bRcV.7 - Question: How were the hyperparameters, such as the 48 anchor points for the UMap and the 0.1 redundancy filter threshold, determined? How sensitive is the final reconstruction performance to these choices?**
> > >
> > > response: As suggested by the reviewer, to assess the sensitivity of our method to the choice of hyperparameters, we conducted additional experiments varying both the number of anchor points in the UMap and the redundancy-filter threshold used during viewpoint selection.
> > >
> > > As for the ablation on the anchor numbers, we refer the reviewer to **3vjJ.2** to see the results and corresponding discussion.
> > >
> > > As for the ablation on the redundancy-filter threshold, this hyperparameter controls how many candidate viewpoints are discarded due to redundancy. A threshold that is too large may filter out excessive viewpoints, while a threshold that is too small may retain too many redundant ones. We evaluate thresholds from 0.0 to 0.3, and the results (**Tab. 5(b)**) show that a threshold of 0.1 yields the best overall performance.
> > >
> > > We have included these ablations and a detailed discussion in line 519 of the revised main text, clarifying the rationale behind our default settings and the robustness of our method with respect to these hyperparameters.

---

### Official Review · Reviewer_yuxC · 2025-10-30

**Soundness:** 2
**Presentation:** 4
**Contribution:** 2
**Rating:** 4
**Confidence:** 4

**Summary:**

The paper proposes a method to address the high computational cost of Active View Selection (AVS) for neural rendering methods. Traditional AVS methods for these models are slow because they require retraining the rendering model at each step to estimate uncertainty.
The paper proposes Peering into the UnkNowN (PUN), a method that uses a lightweight feedforward network, UPNet, to directly predict a neural uncertainty map (UMap) from a single input image. This decouples the view selection process from the costly model retraining.

To train UPNet, the paper proposes generating a Neural Uncertainty Map (NUM) dataset from ShapeNet objects. It takes a single view of an object, trains a single-view synthesis model (Splatter-Image), and then calculates the reconstruction error (e.g., PSNR) between its synthesized images and ground-truth images at 48 fixed "anchor" viewpoints. This 48-dimensional error vector serves as the ground-truth UMap for the initial input view.

At inference time, PUN takes the current view, passes it through the trained UPNet to predict a UMap, and aggregates these maps over time to select the next-best-view with the highest uncertainty, while filtering out redundant views. The paper claims this method is up to 400x faster than baselines, achieves comparable reconstruction accuracy to an upper bound with half the views, and generalizes to novel objects and complex, realistic scenes.

**Strengths:**

1. The paper is very well-written and easy to follow. The overview figures (e.g., Figs. 1 and 2) are high-quality and provide a good overview of the proposed method and task. The structure is logical, and the claims are stated unambiguously.

2. The paper's primary strength is its computational efficiency at inference time. By replacing a full retraining loop with a single forward pass of the lightweight UPNet, the authors achieve a reported 400x speedup in selection time and massive reductions in resource usage.

3. The method empirically outperforms the selected baselines (A-NeRF, NVF, WD)  in reconstruction quality across most metrics on the test datasets (e.g., Table 1).

4. The paper shows surprisingly good zero-shot generalization results, where UPNet, trained only on synthetic ShapeNet objects, is used to select views for complex realistic scenes from NeRFAssets and MIP360, outperforming baselines

**Weaknesses:**

1. The papers premise rests on the proposed ground truth uncertainty maps. However, they do not actually measure uncertainty (as do entropy or variance), but it is a map of the reconstruction error for a specific single-view reconstruction model. I would like to see a more formal analysis of why the model should be an effective policy for guiding a multi-view reconstruction, especially for a completely different model class like NeRF.

2. I also believe that the main claim regarding efficiency is a bit misleading. The stated gains only refer to inference time. It completely ignores the massive offline computational cost required to generate the NUM dataset. The paper states it's impractical to train a model for each viewpoint. However, the solution still requires taking 1300 objects (13 categories x 100 instances), sampling 48 viewpoints for each, running a synthesis model (Splatter-Image) from each view, and generating 48 novel views from each of those. This process, repeated to create 62,400 UMap pairs, represents a colossal, one-time computational burden. The paper does not solve the computational problem; it merely amortizes it.

3. In terms of novelty, I am questioning what we learn from this paper. The UPNet is trained entirely on the ShapeNet dataset. This dataset consists of single, isolated objects on white backgrounds, with views sampled on a sphere. This setup has a significant domain gap compared to any real-world application. It is highly questionable how a model trained on this data can learn a meaningful uncertainty prior for complex, 360-degree, cluttered real-world scenes beyond single objects.

4. I am wondering how this method would compare against an active heuristic-based method, such as, for example, a greedy farthest point sampling instead of simply uniform sampling.

5. The UMap is not a continuous map. It is a fixed 48-dimensional vector corresponding to 48 predefined, fixed relative anchor poses. This is a strong, discretizing assumption that limits the granularity of the uncertainty prediction.

**Questions:**

Can the paper provide a stronger conceptual justification for why predicting the reconstruction error of a single-view 3DGS model is a valid and effective proxy for guiding a multi-view NeRF reconstruction?

Given the massive domain gap between the ShapeNet training data (isolated objects, white background) and the MIP360 test data (complex, cluttered 360 scenes), what features does UPNet actually learn that allow it to generalize so effectively?

How are the ground truth views chosen? Are they always the same 48 poses, or are they variable? Why do we choose 48 views? Why not 24, 64, or some other number? Is this ablated somewhere?

---

> ### Author Response · Authors · 2025-11-21
> **Response part 1**
>
> **yuxC.1 - Weakness: The papers premise rests on the proposed ground truth uncertainty maps. However, they do not actually measure uncertainty (as do entropy or variance), but it is a map of the reconstruction error for a specific single-view reconstruction model. I would like to see a more formal analysis of why the model should be an effective policy for guiding a multi-view reconstruction, especially for a completely different model class like NeRF.**
>
> response: We thank the reviewer for raising this important point. Our work is based on the observation that reconstruction error is a more direct and task-aligned signal for informative viewpoint selection under limited observation cost. Prior AVS work (e.g., [A]) shows that a well-designed uncertainty metric can closely approximate the underlying reconstruction error and thereby effectively guide viewpoint selection. This motivates our approach: instead of relying on handcrafted surrogates, we directly learn to predict this error distribution over views. Regarding why such an error-based map can guide multi-view reconstruction, single-view synthesis models tend to make large errors precisely at viewpoints that are geometrically ambiguous and difficult to reconstruct, and these “hard” viewpoints are common across reconstruction models because all of them optimize the same underlying objective of minimizing reconstruction error. In other words, by learning from many objects, the single-view backbone captures a generic notion of viewpoint difficulty, and these difficult regions remain difficult for multi-view NeRF as well. Therefore, a policy trained to predict such error patterns naturally generalizes and remains effective for guiding viewpoint selection across different reconstruction backbones. We have included this discussion in line 848 of the revised appendix.
>
> **yuxC.2 - Weakness: I also believe that the main claim regarding efficiency is a bit misleading. The stated gains only refer to inference time. It completely ignores the massive offline computational cost required to generate the NUM dataset. The paper states it's impractical to train a model for each viewpoint. However, the solution still requires taking 1300 objects (13 categories x 100 instances), sampling 48 viewpoints for each, running a synthesis model (Splatter-Image) from each view, and generating 48 novel views from each of those. This process, repeated to create 62,400 UMap pairs, represents a colossal, one-time computational burden. The paper does not solve the computational problem; it merely amortizes it.**
>
> response: We thank the reviewer for the thoughtful comment. We agree that generating the NUM dataset involves a one-time offline computational cost. Our main contribution, however, concerns the online efficiency of active view selection. Prior AVS methods require repeatedly training or refining a reconstruction model at every selection step, which makes them impractical for real-time or on-board applications. In contrast, our method replaces this process with a single forward pass of UPNet, offering substantial speedup while maintaining competitive reconstruction quality.
>
> Regarding dataset generation, the cost is incurred once and amortized across all future uses. In practice, using one A6000 GPU, producing each UMap requires about 3 seconds, resulting in roughly 52 hours for the full NUM dataset containing 1300 object instances. For comparison, prior methods require more than 3 hours to select 20 views for a single object, meaning that the same time budget would allow processing only about 14 objects. Although amortization is involved, the overall computational load is still substantially lower when considering repeated deployments of our methods.
>
> We have clarified this line 863 of the revised appendix: our focus is on improving the efficiency of online view selection, while dataset creation is an offline preprocessing step.
>
> [A] Jin L, Chen X, Rückin J, et al. Neu-nbv: Next best view planning using uncertainty estimation in image-based neural rendering[C]//2023 IEEE/RSJ International Conference on Intelligent Robots and Systems (IROS). IEEE, 2023: 11305-11312.

---

> > ### Author Response · Authors · 2025-11-21
> > **Response part 2**
> >
> > **yuxC.3 - Weakness: In terms of novelty, I am questioning what we learn from this paper. The UPNet is trained entirely on the ShapeNet dataset. This dataset consists of single, isolated objects on white backgrounds, with views sampled on a sphere. This setup has a significant domain gap compared to any real-world application. It is highly questionable how a model trained on this data can learn a meaningful uncertainty prior for complex, 360-degree, cluttered real-world scenes beyond single objects.**
> >
> > response: We thank the reviewer for the thoughtful comment. Our work follows the object-level reconstruction setting adopted by most prior AVS literature, as our primary goal is to address the inference-time inefficiency that makes existing methods impractical for online or on-board usage. Within this setting, training UPNet on ShapeNet offers a controlled yet diverse environment that enables us to systematically learn a generic heuristic of viewpoint difficulty across many object categories.
> >
> > Importantly, although UPNet is trained on synthetic object-centric data, our experiments on the real-world MIP360 dataset (**Tab. 3**) show that the learned prior generalizes beyond the training domain. The predicted uncertainty maps remain meaningful in cluttered 360° scenes and continue to guide effective multi-view reconstruction.
> >
> > More broadly, this work provides the first empirical validation that using predicted reconstruction error directly as the signal for active view selection is effective. Across multiple datasets (**Tab.1** and **Tab. 3**) and with different reconstruction backbones (**Tab. 1 ©**), the predicted UMaps consistently lead to strong view-selection performance. This suggests that reconstruction error is a principled and generalizable cue for deciding informative viewpoints under limited observation budgets.
> >
> > Finally, the AVS works in the literature have been focusing on object-level reconstruction. We are adopting their standards for fair comparisons with their models. We agree that scene-level AVS is an important future step. We encourage the community to shift the focus from the standard object-level reconstruction to more complex scene-level reconstruction. We added this in the future work in line 537 of the revised main text.
> >
> >
> > **yuxC.4 - Weakness: I am wondering how this method would compare against an active heuristic-based method, such as, for example, a greedy farthest point sampling instead of simply uniform sampling.**
> >
> > response: As the review suggested, in addition to the uniform policy, we conducted experiments using a greedy farthest point sampling (FPS) strategy as the active heuristic. The results, provided in **Tab. S6** in the revised version, show that our method continues to achieve competitive performance compared with FPS. We also note that, in an ideal setting, FPS tends to produce a viewpoint distribution similar to uniform sampling—both aim to maximize pairwise angular separation. For this reason, we refer the reviewer to **Tab. S1**, which quantitatively compares our method with the uniform sampling policy. As shown, both FPS and uniform sampling achieve strong results on the visual coverage metric, since this metric directly reflects the spatial spread of the selected viewpoints. And our method outperforms both of these methods.

---

> > > ### Author Response · Authors · 2025-11-21
> > > **response part 3**
> > >
> > > **yuxC.5 - Weakness: The UMap is not a continuous map. It is a fixed 48-dimensional vector corresponding to 48 predefined, fixed relative anchor poses. This is a strong, discretizing assumption that limits the granularity of the uncertainty prediction.**
> > >
> > > response: We appreciate the reviewer’s observation. The UMap is indeed defined over 48 anchor viewpoints, and this discrete formulation reflects a practical trade-off. Increasing the number of anchors would substantially raise the computational cost of dataset generation, since each anchor requires running a single-view synthesis model to obtain its reconstruction error. It would also make inference more expensive due to the need for denser interpolation across a larger set of anchors. Moreover, as in many computational representations—such as pixels for images or grids for 3D maps—a certain degree of discretization is both necessary and beneficial for providing a stable, learnable approximation of a continuous space. Our choice of 48 anchors offers a balance between angular granularity and computational feasibility.
> > >
> > > We also refer the reviewer to the ablation results (**Tab. 5(a)**), where we vary the number of anchor viewpoints to examine how different UMap densities influence downstream performance. The results show that increasing the number of anchors from 12 → 48 yields a clear performance gain across most metrics, indicating that denser angular sampling indeed improves the fidelity of the UMap. However, further increasing the count from 48 → 108 results in only marginal improvements while the time required to generate the NUM dataset more than doubles. This diminishing return suggests that 48 anchors offer a good balance between reconstruction performance and computational cost. These results have been included and discussed in the line 511 of the revised main text.
> > >
> > > **yuxC.6 - Question: Can the paper provide a stronger conceptual justification for why predicting the reconstruction error of a single-view 3DGS model is a valid and effective proxy for guiding a multi-view NeRF reconstruction?**
> > >
> > > response: We thank the reviewer for the thoughtful question. Conceptually, predicting the reconstruction error of a single-view 3DGS model is effective because reconstruction difficulty is largely determined by the geometry and visibility structure of the object, rather than the specific reconstruction backbone. Viewpoints that are occluded or geometrically complex tend to produce higher errors for any reconstruction model—single-view or multi-view—because all such models ultimately optimize the same objective of minimizing view-dependent reconstruction error. As a result, the error patterns produced by a single-view 3DGS predictor reflect a general notion of “viewpoint difficulty” that is shared across model classes. Please refer to **yuxC.7** for a more detailed discussion.
> > >
> > > By training UPNet to approximate this error landscape across many objects, the network learns to identify viewpoints that are inherently hard or easy to reconstruct, independent of the downstream backbone. This explains why the predicted UMaps remain effective for guiding a multi-view NeRF reconstruction: although NeRF is a different model class, the underlying sources of error (occlusion, lack of visibility) are the same. Our experiments across datasets and reconstruction backbones empirically confirm this consistency. We have clarified this conceptual connection more explicitly in line 1113 of the revised appendix.

---

> > > > ### Author Response · Authors · 2025-11-21
> > > > **Response part 4**
> > > >
> > > > **yuxC.7 - Question: Given the massive domain gap between the ShapeNet training data (isolated objects, white background) and the MIP360 test data (complex, cluttered 360 scenes), what features does UPNet actually learn that allow it to generalize so effectively?**
> > > >
> > > > response: We thank the reviewer for the thoughtful question. To better understand what UPNet learns, we further examine whether basic geometric and texture–complexity indicators correlate with the reconstruction error signal used in our UMap supervision. For each anchor viewpoint, we compute several standard view-complexity measures: (1) depth_grad_var: variance of depth gradients, capturing geometric relief and surface discontinuities; (2) edge_density: proportion of high-gradient pixels in the depth image, reflecting structural detail; (3) color_grad: magnitude of color gradients, indicating texture richness; (4) color_entropy: entropy of the color histogram, measuring appearance diversity; (5) laplacian_energy: Laplacian magnitude, capturing high-frequency texture and edges. We refer the reviewer to the line 1113 of the revised appendix for a detailed introduction of these complexity metrics.
> > > >
> > > > As shown in **Tab. S7** in the revised appendix, all metrics show moderate correlation, indicating that viewpoints with greater geometric relief, higher edge density, or richer texture tend to exhibit higher reconstruction error for the single-view synthesis backbone. We also visualize the scatter plot between the complexity metrics and reconstruction error metrics in **Fig. S7** to provide a better understanding of this correlation.
> > > >
> > > > By learning such patterns over a large number of diverse objects in ShapeNet, UPNet acquires a generic notion of viewpoint difficulty that is largely domain-agnostic.
> > > >
> > > > Empirically, this is supported by our experiments on the MIP360 dataset (**Tab. 3**), where UPNet—trained solely on synthetic, object-centric data—still produces meaningful UMaps and guides competitive multi-view reconstruction performance. This indicates that the model does not rely on dataset-specific appearance cues, but instead captures geometry and texture complexities that remain valid in real 360° environments.
> > > >
> > > > We have included the results and this insight in line 1113 of the revised appendix.
> > > >
> > > > **yuxC.8 - Question: How are the ground truth views chosen? Are they always the same 48 poses, or are they variable? Why do we choose 48 views? Why not 24, 64, or some other number? Is this ablated somewhere?**
> > > >
> > > > response: We thank the reviewer for the helpful question. The ground-truth views used to construct the uncertainty maps correspond to the anchor viewpoints, which are designed to be uniformly distributed on the sphere surface in order to provide a complete and balanced coverage of possible viewing directions. These anchor viewpoints maintain fixed relative positions with respect to the input view, as described in line 194 of the revised main text. HealPix supports resolution levels via the parameter $n_{side}$ and the total pixel count is given by $N=12\times{n_{side}}^2$. For example, $n_{side} = 1$ gives 12 pixels, $n_{side} = 2$gives 48 pixels. We chose 48 as it offers a practical balance between angular granularity and dataset‐generation cost.
> > > >
> > > > To address the reviewer’s question regarding anchor count, we conducted an ablation across different numbers of anchors. As shown in **Tab. 5 (a)** of the revised main text, increasing the number of anchors from 12 → 48 yields a clear performance gain across most metrics, indicating that denser angular sampling indeed improves the fidelity of the UMap. However, further increasing the count from 48 → 108 results in only marginal improvements while the time required to generate the NUM dataset more than doubles. This diminishing return suggests that 48 anchors offer a good balance between reconstruction performance and computational cost.

---

> > > > > ### Comment · Reviewer_yuxC · 2025-11-26
> > > > > **Thank you for the clear and detailed rebuttal**
> > > > >
> > > > > I would like to thank the authors on providing a thorough and detailed rebuttal. I appreciate the effort you put into conducting the additional experiments and providing the clarifications, which address many of the core issues raised by the my colleagues and me.
> > > > >
> > > > > In particular, I found the following points highly valuable:
> > > > > * The clarification that single-view reconstruction error provides a domain-agnostic heuristic for "viewpoint difficulty," rooted in generic geometric and visibility structure, is conceptually strong. This is supported by the new experiment showing effectiveness when trained on a NeRF-based NUM and the correlation analysis with complexity metrics.
> > > > >
> > > > > * The analysis of the features learned by UPNet, correlating UMap predictions with geometric and texture complexity indicators, provides the necessary insight into the strong zero-shot generalization to real-world scenes like MIP360.
> > > > >
> > > > > * I accept the clarification that the primary contribution is the online efficiency (400x speedup), and that the NUM dataset creation is an amortized, one-time offline cost. As long as this is mentioned as such in the paper, I am ok with this.
> > > > >
> > > > > However, while my confidence in the submission has increased and my major concerns regarding soundness and generalization have been largely resolved, I remain somewhat cautious about fully embracing the contribution at the highest level. The massive, one-time offline cost, although amortized, is still a significant practical hurdle compared to fully online methods, even if the online speedup is undeniable.
> > > > >
> > > > > Given the significant improvements and the strong experimental results across datasets and backbones, the paper is clearly above the acceptance threshold for me.
> > > > >
> > > > > I am therefore increasing my rating from 4  to 6.

---

> > > > > > ### Author Response · Authors · 2025-11-27
> > > > > >
> > > > > > Thank you for the constructive comments and for carefully reviewing our rebuttal and additional experiments. We will incorporate all of your valuable suggestions into the revised version to ensure the final manuscript clearly conveys these points. Please feel free to let us know if anything else remains unclear.

---

### Official Review · Reviewer_3VjJ · 2025-10-31

**Soundness:** 3
**Presentation:** 3
**Contribution:** 3
**Rating:** 6
**Confidence:** 4

**Summary:**

The paper tackles next-best-view selection for 3D reconstruction. The authors fine-tune a ViT to predict, from a single input image, a reconstruction uncertainty/score map. Supervision for fine-tuning is derived from PSNR, SSIM, LPIPS, or MSE computed on outputs of Splatter-Image, a monocular 3D reconstruction model; training data is curated from ShapeNet. At inference, the next best view is selected sequentially: uncertainties at each candidate view are interpolated from a fixed set of spherical anchors and aggregated across timesteps. Experiments on synthetic and real scenes show the method achieves comparable reconstruction quality with fewer views and lower computation, and the selection generalizes to different reconstruction backbones (e.g., 3DGS).

**Strengths:**

* Originality: Next-best-view selection is a classic problem in 3D vision; the paper’s backbone-agnostic approach—predicting a feed-forward uncertainty/score map for view selection—is a novel angle within this space.

* Quality:

  * The solution is simple and effective.
  * The evaluation is comprehensive, covering both synthetic and real scenes, with robustness tests (e.g., lighting and distance). Generalization to multiple reconstruction backbones is also tested and supports the claim.

* Clarity:

  * The paper is well organized and clearly written, especially in framing the problem and motivation.
  * The proposed method is introduced clearly, with the selection pipeline easy to follow.

* Significance:

  * The approach improves efficiency of existing 3D reconstruction pipelines while maintaining comparable performance.
  * Backbone agnosticism enables plug-and-play use in arbitrary pipelines.
  * The method’s simplicity makes it easy to extend for follow-up research.

**Weaknesses:**

* Uncertainty definition: The paper uses PSNR/SSIM/LPIPS/MSE as “uncertainty” labels; all are photometric and ignore geometry quality. This limits relevance for downstream tasks that depend on accurate shape.

* Anchor design lacks justification: The anchor layout (48 HEALPix points) is fixed without analysis; it is unclear whether performance is bottlenecked by anchor density or discretization choice.

* Candidate sampling is unclear: “Randomly sample 512 candidates” is under-specified (sphere-uniform vs. azimuth/elevation uniform vs. more sophisticated sampling). The chosen distribution can materially affect results.

* Scope beyond object-centric canonical sphere: Training/evaluation assume canonical camera shells around single objects; performance and assumptions may not transfer to indoor/outdoor scenes with general camera placement, depth variation, and occlusions.

* Single-image conditioning vs. relative view dependency: The ViT takes only one image, yet the output implicitly depends on the relative pose between the input view and the 48 anchors; the current design does not encode this dependency explicitly.

**Questions:**

- First, please refer to the weakness.
- Second, are there cases where high uncertainty ≠ reconstruction gain?
- Third, does the model has explicit pose conditioning and why?

---

> ### Author Response · Authors · 2025-11-21
> **Response part 1**
>
> **3VjJ.1 - Weakness: Uncertainty definition: The paper uses PSNR/SSIM/LPIPS/MSE as “uncertainty” labels; all are photometric and ignore geometry quality. This limits relevance for downstream tasks that depend on accurate shape.**
>
> response: While our current UMap labels are based on photometric reconstruction error (PSNR/SSIM/LPIPS/MSE), we would like to clarify that this choice does not prevent the method from achieving strong geometric performance. As shown in **Tab. S1**, even without explicitly modeling geometric uncertainty, our method achieves competitive results on geometry-based metrics such as Acc and CR, indicating that photometric error serves as a useful proxy for geometric reconstruction difficulty in practice.
>
> There are also two practical reasons why we did not use geometry quality as supervision in this work. First, generating geometry-based NUM labels is computationally expensive. It requires reconstructing meshes and comparing them with ground-truth surfaces, which involves heavy distance-field or visibility-aware computations. Second, extracting geometry directly from the single-view synthesis backbone is often unstable and inaccurate, which would cause the supervision signal to be dominated by the artifacts of the geometry-extraction method rather than the underlying reconstruction difficulty. Incorporating geometric error would therefore require its own optimization pipeline, which is beyond the scope of this work.
>
> That said, we fully agree with the reviewer that geometry-aware uncertainty is an important direction for future research, and we believe our framework can naturally incorporate such signals once accurate and efficient geometric supervision becomes available. We have clarified this discussion in the line 227 of the revised main text.
>
>
> **3VjJ.2 - Weakness: Anchor design lacks justification: The anchor layout (48 HEALPix points) is fixed without analysis; it is unclear whether performance is bottlenecked by anchor density or discretization choice.**
>
> response: The anchor viewpoints used in our UMap are generated using the HEALPix uniform-sphere sampling scheme. HEALPix provides only discrete numbers of uniformly distributed directions, determined by the resolution parameter $n_{side}$, where the total number of anchor points is $N = 12\times {n_{side}}^2$.
>
> Thus, the allowable uniform anchor counts are 12, 48, 108, etc., corresponding respectively to n_side = 1, 2, 3. We choose 48 anchors because it provides a practical balance between angular granularity and the computational cost of generating the NUM dataset. Increasing the anchor count substantially raises data-generation time and memory requirements, while fewer anchors may provide insufficient directional resolution for stable UMap prediction.
>
> We have conducted an ablation study across multiple HEALPix resolutions (12, 48, and 108 anchors) to analyze how anchor density affects UMap quality and downstream view-selection performance. As shown in the **Tab. 5 (a)**, increasing the number of anchors from 12 → 48 yields a clear performance gain across most metrics, indicating that denser angular sampling indeed improves the fidelity of the UMap. However, further increasing the count from 48 → 108 results in only marginal improvements while the time required to generate the NUM dataset more than doubles (from 135 seconds to 281 seconds per viewpoint). This diminishing return suggests that 48 anchors offer a good balance between reconstruction performance and computational cost. We have included these results in the **Tab. 5 (a)** and the accompanying discussion in the line 511 of the revised main text.
>
> **3VjJ.3 - Weakness: Candidate sampling is unclear: “Randomly sample 512 candidates” is under-specified (sphere-uniform vs. azimuth/elevation uniform vs. more sophisticated sampling). The chosen distribution can materially affect results.**
>
> response: We would like to clarify this point as follows. Our candidate viewpoints are generated by uniformly sampling 512 azimuth angles and 512 elevation angles over the entire sphere, and forming viewing directions from their Cartesian combinations. This produces a dense and approximately uniform coverage of the full spherical viewpoint space used in our experiments. We agree that the sampling strategy should be described more clearly, and we have included a detailed explanation in line 241 of the revised main text.

---

> > ### Author Response · Authors · 2025-11-21
> > **Response part 2**
> >
> > **3VjJ.4 - Weakness: Scope beyond object-centric canonical sphere: Training/evaluation assume canonical camera shells around single objects; performance and assumptions may not transfer to indoor/outdoor scenes with general camera placement, depth variation, and occlusions.**
> >
> > response: We would like to clarify this point as follows. While our framework is trained under the canonical object-centric setting commonly adopted in prior AVS work, we conducted three evaluations that directly examine its behavior beyond this simplified setup. First, as shown in our experiments on the real-world MIP360 dataset (**Tab. 2**), UPNet—trained solely on synthetic ShapeNet objects—still produces meaningful UMaps and guides competitive multi-view reconstruction in cluttered, full-scene 360° environments. Second, our camera-distance variation experiments (**Tab. 1 (d)**) show that UPNet remains stable across different viewing radius, demonstrating robustness to depth variation. Third, as noted in our response to **Wvnw.3**, the experiments with targets shifted away from the image center (**Tab. S8**) show that UPNet remains stable under moderate object displacements. In summary, these evaluations indicate that the learned prior generalizes beyond the object-centric, canonical-sphere setting.
> >
> > These observations align with our broader finding that UPNet learns a generic heuristic of viewpoint difficulty —driven primarily by geometric and visibility complexities—which enables transfer to indoor and outdoor scenes. We refer the reviewers to our response to **yuxC.7** and results in **Tab. S7** and **Fig. S7** for a more detailed analysis.
> >
> > That said, we agree with the reviewer that scene-level AVS with free-space camera motion and complex occlusion patterns is an important next step. Extending UPNet to scene-level AVS is a meaningful future research direction, and we have included this in line 539 of the revised main text.
> >
> >
> > **3VjJ.5 - Weakness: Single-image conditioning vs. relative view dependency: The ViT takes only one image, yet the output implicitly depends on the relative pose between the input view and the 48 anchors; the current design does not encode this dependency explicitly.**
> >
> > response: We clarify this concern below. In UPNet, the model does not need to explicitly encode the relative pose between the input view and the 48 anchors. This is because the anchors are defined in a fixed canonical coordinate frame relative to the input viewpoint. In other words, UPNet always predicts uncertainty values for 48 viewpoints that lie at fixed relative directions w.r.t. the input camera.
> >
> > Given this design, the dependency between the input view and each anchor is implicitly handled by the deterministic geometric transformation used at inference time: once the input camera pose is known, the global pose of each anchor can be computed analytically by applying the fixed relative offsets. As a result, UPNet only needs to output uncertainty scores in the canonical anchor frame; the conversion to world coordinates happens outside the network.
> >
> > Thus, the model does not require additional pose encoding to reason about anchor-relative configurations. The relative-view dependency is already embedded in the fixed anchor layout and the deterministic transformation applied at inference time. We have clarified this design in line 195 of the revised main text.
> >
> >
> > **3VjJ.6 - Question: First, please refer to the weakness.**
> >
> > response: We believe our response to the corresponding weakness addresses this concern, and we hope it sufficiently resolves the reviewer’s point.
> >
> >
> > **3VjJ.7 - Question: Second, are there cases where high uncertainty ≠ reconstruction gain?**
> >
> > response: We thank the reviewer for the question. In our framework, the uncertainty metric is defined directly as the reconstruction error, so under accurate prediction, high uncertainty corresponds to high potential reconstruction gain by design. As shown in **Fig. 3** and discussed in line 417 of the revised main text, the predicted UMaps exhibit a strong correlation with the ground-truth reconstruction error (Pearson correlation 0.82), indicating that UPNet is able to approximate the reconstruction-error landscape reliably.
> >
> > We have included **Fig. S4** in the revised, which further illustrates the close alignment between predicted and ground-truth uncertainties.. The scatter plot shows a clear linear trend (red line), indicating a high correlation between the two and demonstrating that UPNet accurately captures the underlying reconstruction-error landscape. These results suggest that, in practice, high predicted uncertainty is a meaningful indicator of expected reconstruction gain.

---

> ### Author Response · Authors · 2025-11-21
> **Response part 3**
>
> **3VjJ.8 - Question: Third, does the model has explicit pose conditioning and why?**
>
> response:  As clarified in our response to **3VjJ.5**, UPNet does not require explicit pose conditioning because the anchors are defined in a fixed canonical coordinate frame relative to the input viewpoint, as illustrated in **Fig. 1(b)** and described in line 195 of the revised main text. The model always predicts uncertainty values for 48 viewpoints that lie at predetermined relative directions with respect to the input camera. Although UPNet does not take the absolute camera pose as input, the global positions of all anchors can be computed analytically from the known input camera pose and the fixed relative offsets. This enables us to use the predicted uncertainties at those anchor positions to guide viewpoint selection without embedding explicit pose information inside the network.

---

### Official Review · Reviewer_Wvnw · 2025-11-02

**Soundness:** 3
**Presentation:** 3
**Contribution:** 1
**Rating:** 4
**Confidence:** 4

**Summary:**

The paper tackles AVS for 3D reconstruction by predicting a neural uncertainty map from a single current view. A lightweight feed-forward network, UPNet, takes one image and outputs uncertainty values over all candidate viewpoints. Using the selected views, standard neural rendering models (NeRF/3DGS) are trained and evaluated against competing AVS methods. The approach attains comparable reconstruction accuracy while dramatically reducing computational cost with up to 400× speedup and >50% lower CPU/RAM/GPU usage. It can also generalize to novel object categories without extra training.

**Strengths:**

This idea is very interesting: training a feed-forward uncertainty prediction network on a self-constructed dataset with ground truth supervision for AVS task. This can greatly reduce the computational resources and time required for next-best-view selection of AVS task. Experiments show that, compared with baseline AVS methods, the proposed approach exhibits clear improvements.

**Weaknesses:**

This paper proposes an interesting and effective method for the AVS task. However, my main concerns are the experimental evaluation and the value of this task. I find it hard to imagine a scenario where, given a single input view, the system should output where the next view should be obtained. For robotic applications, if it has the ability to move from one viewpoint to another, then it can capture dense views at arbitrary positions. For multi-view reconstruction, all ground truth captured views are already provided, and there is no need to select a subset for training.

If the authors claim that the goal of this task is to reduce the number of training views so as to reduce training time while maintaining quality, then I believe the evaluation in this work is incomplete. At a minimum, it should include comparisons of rendering quality and training resources with the following: (1) 3DGS trained on all available views. (2) pipelines of sparse-view NVS 3DGS (e.g. FSGS, Binocular3DGS) based on the selected small set of views.

**Questions:**

1. In the experimental setup, the authors removed real-scene views that are not centered on the object. In practical applications, what happens if the initial view is not oriented toward the object center?

2. The UPNet gives sampling quality at 48 uniformly distributed positions on the sphere. However, in real scenes, a full 360-degree range of view may not be available, for example, some of the views may be under the ground (MipNeRF360 dataset). If the next best view is predicted in such a region, how should it be handled?

---

> ### Author Response · Authors · 2025-11-21
> **response to weakness**
>
> **Wvnw.1 - Weakness: This paper proposes an interesting and effective method for the AVS task. However, my main concerns are the experimental evaluation and the value of this task. I find it hard to imagine a scenario where, given a single input view, the system should output where the next view should be obtained. For robotic applications, if it has the ability to move from one viewpoint to another, then it can capture dense views at arbitrary positions. For multi-view reconstruction, all ground truth captured views are already provided, and there is no need to select a subset for training.**
>
> response:  We appreciate the reviewer’s comment and would like to clarify the concern as follows. First, our setting follows the established evaluation protocol adopted by prior AVS studies, where the system observes only a single initial view and must actively decide subsequent viewpoints under a fixed and inherently limited observation budget. Second, the core value of AVS lies in selecting informative viewpoints when the observation cost—not the training cost—is the primary constraint. In many real scenarios such as robotic inspection, cultural heritage digitization, and autonomous exploration, an agent cannot afford to capture arbitrarily dense views due to physical and temporal limitations (e.g., battery life, motion time, or mission duration). Under this assumption, our method does not select a subset from densely captured views; instead, it actively selects the most informative next viewpoint based solely on the currently observed data. We have update line 49 of the Introduction to provide a clearer motivation for the AVS task and to explicitly articulate its realistic application scenarios in practical 3D data acquisition.
>
> **Wvnw.2 - Weakness: If the authors claim that the goal of this task is to reduce the number of training views so as to reduce training time while maintaining quality, then I believe the evaluation in this work is incomplete. At a minimum, it should include comparisons of rendering quality and training resources with the following: (1) 3DGS trained on all available views. (2) pipelines of sparse-view NVS 3DGS (e.g. FSGS, Binocular3DGS) based on the selected small set of views.**
>
> response: We thank the reviewer for the constructive suggestions. We would like to clarify that the goal of our work is not to reduce the training time of the reconstruction backbone, but to achieve the best possible reconstruction quality under a limited observation budget. Therefore, the key efficiency concern lies in the view-selection stage, while the training resources of the reconstruction backbone are not a decisive factor for comparison, as all view-selection methods are evaluated using the same reconstruction backbone under identical training configurations.
>
> Regarding the reviewer’s first suggestion, we note that **Tab. 1** already includes a baseline trained with a larger number of views (the upper-bnd setting). This evaluation was performed using NeRF as the reconstruction backbone. In addition, we conducted the same experiment using 3DGS as the backbone; the results are provided in **Tab. S1** in the main text. Consistent with using NeRF as the reconstruction backbone, the upper-bnd achieves the best performance and our method achieves performance that is the closest to the upper-bnd.
>
> For the second suggestion, we further evaluated all methods using Binocular3DGS as the reconstruction backbone and compared the reconstruction quality obtained from viewpoints selected by our method vs. NVF. The results, shown in **Tab. 2**, again demonstrate that our approach outperforms NVF across all metrics. Since all methods share the same reconstruction pipeline and hyperparameters, the training resources remain identical across different AVS methods. The discussion is included in line 372 of the revised main text.

---

> > ### Author Response · Authors · 2025-11-21
> > **response to the questions**
> >
> > **Wvnw.3 - Question: In the experimental setup, the authors removed real-scene views that are not centered on the object. In practical applications, what happens if the initial view is not oriented toward the object center?**
> >
> > response: We thank the reviewer for the insightful question. We acknowledge that assuming the target to be roughly centered in the scene is a limitation of the current setting. In practice, our method exhibits a certain degree of tolerance to small spatial offsets. For instance, our experiments on the MIP360 dataset already contain examples where the object is not perfectly centered, and the method remains effective. In addition, we conducted further evaluations under the novel-category setting by deliberately shifting the object by a small amount in 3D space (e.g., ∆xyz = (0, 0, 0.2)). The results in **Tab. S8** in the revised version, show that our method still achieves competitive performance in this case. We have clarified this tolerance to moderate misalignment in line 1017 of the revised appendix.
> >
> > However, when the spatial offset between the camera and the object becomes very large, our current setup cannot reliably establish the correspondence between the predicted U-Map and the true viewing directions, making next-view selection less reliable. We emphasize that this is not a technical limitation of our framework. In principle, one could generate U-Maps under arbitrary camera poses that are not centered on the target object and subsequently train UPNet on such pose-augmented datasets. This is entirely feasible; the main challenge lies in the substantial computational cost of densely sampling and augmenting camera poses in 3D space, which would require generating a significantly larger NUM dataset. For this reason, in the current paper we focus on the canonical spherical setting and provide a proof-of-concept demonstration of our approach. Extending UPNet to free-space camera placement remains an important direction for future work.
> >
> > **Wvnw.4 - Question: The UPNet gives sampling quality at 48 uniformly distributed positions on the sphere. However, in real scenes, a full 360-degree range of view may not be available, for example, some of the views may be under the ground (MipNeRF360 dataset). If the next best view is predicted in such a region, how should it be handled?**
> > response: We thank the reviewer for raising this practical question. This issue is directly related to our assumption about the candidate viewpoints. In our experiments on ShapeNet-style object scenes, we assume a full 360-degree spherical viewpoint space, since the objects are captured in isolation. For realistic datasets, such as MipNeRF360, the candidate viewpoints are not defined over the full sphere but are instead restricted to the set of valid camera poses provided by the dataset. Our method always performs selection within this feasible set.
> >
> > Therefore, if certain viewpoints are physically impossible—such as viewpoints below the ground plane or occluded by obstacles—they are simply excluded from the candidate viewpoints, and the algorithm never considers them. In other words, UPNet predicts uncertainty for all anchor directions, but the final next-view decision is made only over the valid subset of candidate viewpoints. We have addressed this point in line 244 of revised main text.

---

> ### Comment · Reviewer_Wvnw · 2025-11-22
>
> Regarding the experiments on sparse-view NVS 3DGS, the original intention behind my comment is the following: current SOTA sparse-view reconstruction methods are already carefully designed for sparse-input settings, so I worry that this might diminish or obscure the benefits brought by different AVS strategies. Fortunately, your newly added experiment in Tab. 2 directly addresses this concern. It shows that even when using a SOTA sparse-view 3DGS backbone, your AVS method still appears supeirority. I encourage the author to integrate this conclusion into the paper, so that readers can better understand why this comparison is important and how it supports your claims.
>
> I also appreciate the newly added experiment in Tab. S8 that analyzes the limitation in terms of the initial view. It is a meaningful and important direction for future work that improving the training strategy of UPNet to better handle off-center viewpoints during generalization.
>
> Given that my concerns have been fully addressed, I am willing to raise my rating from 4 to 8.

---

> > ### Author Response · Authors · 2025-11-22
> >
> > Thank you for your valuable feedback. We will definitely integrate these points into the revised paper. Please feel free to let us know if anything else remains unclear.

---

### Author Response · Authors · 2025-11-30
**Summary**

Across all four reviewers, the main concerns converged on three aspects. First, reviewers questioned the soundness of using single-view reconstruction error as the supervisory signal for UMaps—specifically whether such supervision captures a generalizable notion of viewpoint difficulty that transfers across reconstruction backbones and to real-world domains. Second, several design choices were viewed as insufficiently justified, including the use of multiplicative versus additive aggregation, the number of anchor viewpoints, and the redundancy threshold. Third, reviewers raised questions about the practical value of the proposed AVS framework, particularly in the presence of strong sparse-view NVS models, the large offline cost required to generate the NUM dataset, and the real-world applicability of a method trained under canonical object-centric spherical settings.

To address the first concern, we performed additional analyses (Tab. S7, Fig. S7) showing that reconstruction error correlates consistently with geometric and texture–complexity metrics, demonstrating that UPNet learns a domain-agnostic heuristic of viewpoint difficulty rather than artifacts of a specific synthesis model. We further provided empirical evidence through experiments on real-world MIP360 scenes (Tab. 3) and through the NUM-NeRF experiment (Tab. S2) to validate this soundness.

For the second concern, we added ablation studies on the additive aggregation policy (Tab. S9), the number of anchor viewpoints (Tab. 5(a)), and the redundancy-filter threshold (Tab. 5(b)). These experiments clarify the rationale behind each design choice and show that our default settings strike a good balance between performance and computational cost.

Regarding the third concern, we conducted additional experiments using a SOTA sparse-view NVS backbone, Binocular3DGS (Tab. 2), demonstrating that our AVS strategy continues to provide clear improvements even when combined with such pipelines. We also added evaluations in which the object is shifted away from the image center (Tab. S8), addressing questions about practical robustness, and clarified that our primary contribution is the large reduction in online inference cost, whereas the NUM-generation cost is a one-time, amortized offline preprocessing step.

**The score changes and the motivations.**
Reviewer Wvnw noted that the newly added Binocular3DGS experiment and the off-center viewpoint study directly resolved their core concerns about the practical value of AVS under sparse-view NVS backbones. After confirming that the method remains effective even when strong sparse-view reconstruction models are used, the reviewer stated that their concerns were “fully addressed” and increased the rating from 4 to 8.
Reviewer yuxC considered several additions “highly valuable,” including the clarification that single-view reconstruction error provides a domain-agnostic prior for viewpoint difficulty, the complexity-correlation analysis explaining UPNet’s generalization, and the explanation of the amortized nature of the NUM dataset cost. Although remaining cautious about the one-time offline cost, the reviewer stated that the paper is clearly above the acceptance threshold and raised the rating from 4 to 6.

---

### Meta-Review · Area_Chair_8raj · 2025-12-24

**Summary:**

The paper proposes a novel Active View Selection (AVS) framework that utilizes a lightweight network (UPNet) to predict uncertainty maps from a single view, significantly reducing online inference costs compared to iterative approaches. The reviewers' initial concerns centered on three main areas: 1) the soundness and generalization of using single-view reconstruction error as a supervisory signal for "uncertainty" (Reviewers yuxC, 3VjJ, bRcV); 2) the practical value and comparison against state-of-the-art sparse-view NVS backbones like Binocular3DGS (Reviewer Wvnw); and 3) the justification for specific design choices such as anchor density and aggregation strategies (Reviewers 3VjJ, bRcV).

The authors provided a robust rebuttal. They demonstrated that their uncertainty metric correlates with geometric complexity (addressing soundness), provided new experiments with Binocular3DGS (addressing practical value), and included ablation studies for design choices. Consequently, the consensus shifted positively, recognizing the method's efficiency and cross-domain generalization capabilities.

**Reviewer Concerns:**

**Addressed Concerns:**

1. Comparison with SOTA Sparse-View Models: Reviewer Wvnw's concern regarding the method's value in the presence of strong baselines was fully addressed by the new experiments with Binocular3DGS (Tab. 2), showing the method remains superior.

2. Soundness of Uncertainty Proxy: Reviewer yuxC and bRcV questioned if the reconstruction error is a valid proxy for uncertainty. The authors effectively addressed this by showing correlations with geometric/texture complexity (Tab. S7) and validating the method using a NeRF-based NUM (Tab. S2), convincing Reviewer yuxC that the heuristic is domain-agnostic.

3. Design Justifications: Concerns from 3VjJ regarding anchor sampling and bRcV regarding aggregation strategies were addressed through targeted ablations (Tab. 5 and Tab. S9), clarifying the trade-offs between performance and computational cost.
Off-Center Robustness: Reviewer Wvnw's question about off-center viewpoints was resolved by the additional analysis in Tab. S8.

**Outstanding Concerns:**

1. Offline Computational Cost: While Reviewer yuxC accepted the "amortization" argument, they noted that the "massive, one-time offline cost" for generating the NUM dataset remains a practical hurdle compared to fully online methods.

2. Scene-Level Generalization: Reviewers 3VjJ and yuxC noted that while object-centric generalization is good, the extension to complex, free-space scene-level AVS remains a future direction rather than a solved problem in this work.

**Reviewer Scores:**

- Reviewer Wvnw: (4 -> 8). The reviewer explicitly stated their concerns were "fully addressed" by the Binocular3DGS and off-center experiments.
- Reviewer yuxC:  (4 -> 6). The reviewer found the complexity-correlation analysis "highly valuable" and raised their score, though remained cautious about the offline costs.
- Reviewer 3VjJ: (6). The reviewer started with a positive score. The clarifications on anchor design (Tab. 5a) and geometric proxies likely solidified this assessment, as no major objections remained.
- Reviewer bRcV: (6). The reviewer started with a positive score. The rebuttal addressing the aggregation strategy (Tab. S9) and the "ground truth" dependence (Tab. S2) effectively defended the initial positive rating.

---

### Decision · Program_Chairs · 2026-01-26

Accept (Poster)